# Genomic determinants of antigen expression hierarchy in African trypanosomes

Zhibek Keneskhanova[1,2,6], Kirsty R. McWilliam[1,2,5,6], Raúl O. Cosentino[1,2,6], Anna Barcons-Simon[1,2], Atai Dobrynin[2,3], Jaclyn E. Smith[4], Ines Subota[1,2], Monica R. Mugnier[4], Maria Colomé-Tatché[2,3✉] & T. Nicolai Siegel[1,2✉]

Antigenic variation is an immune evasion strategy used by many different pathogens. It involves the periodic, non-random switch in the expression of different antigens throughout an infection. How the observed hierarchy in antigen expression is achieved has remained a mystery[1,2]. A key challenge in uncovering this process has been the inability to track transcriptome changes and potential genomic rearrangements in individual cells during a switch event. Here we report the establishment of a highly sensitive single-cell RNA sequencing approach for the model protozoan parasite *Trypanosoma brucei*. This approach has revealed genomic rearrangements that occur in individual cells during a switch event. Our data show that following a double-strand break in the transcribed antigen-coding gene—an important trigger for antigen switching—the type of repair mechanism and the resultant antigen expression depend on the availability of a homologous repair template in the genome. When such a template was available, repair proceeded through segmental gene conversion, creating new, mosaic antigen-coding genes. Conversely, in the absence of a suitable template, a telomere-adjacent antigen-coding gene from a different part of the genome was activated by break-induced replication. Our results show the critical role of repair sequence availability in the antigen selection mechanism. Furthermore, our study demonstrates the power of highly sensitive single-cell RNA sequencing methods in detecting genomic rearrangements that drive transcriptional changes at the single-cell level.

A common strategy used by pathogens to evade the host immune response is antigenic variation. Antigenic variation refers to the ability of a pathogen to systematically alter the expression of an antigen on its surface to evade identification and subsequent elimination by the host's immune response[1]. This strategy is evident in a wide range of evolutionarily distant pathogens, particularly so in *Trypanosoma brucei*[2]. *T. brucei* is a single-celled parasite that lives extracellularly in the bloodstream, adipose tissue[3] and skin[4] of its mammalian host and is transmitted by tsetse flies. It is the causative agent of the neglected tropical disease human African trypanosomiasis and the wasting disease nagana in animals. More than $10^7$ identical variant surface glycoproteins (VSG) form a dense, homogeneous surface coat enshrouding the trypanosome. *T. brucei* is able to evade antibody-mediated clearance from the mammalian bloodstream by antigenic variation of this highly immunogenic VSG coat to antigenically distinct VSG isoforms.

The trypanosome genome contains roughly 2,500 distinct VSG genes, most of which are archived in silent subtelomeric arrays in the 11 diploid megabase chromosomes[5]. Most of the remaining VSG genes are located within a highly specialized repertoire of roughly 100 30–150-kb minichromosomes. Minichromosomes contain a 20–80-kb core of 177-bp

repeats and one or two VSG genes at their subtelomeres[6,7]. Expression of a VSG relies on the positioning of a VSG gene within one of roughly 15 telomere-proximal bloodstream expression sites (BES), which are subject to strict mutually exclusive expression, ensuring that only one is kept transcriptionally active at a time. The general framework of BESs is highly conserved[8,9]. Following an RNA polymerase I promoter, the BES contains up to 11 expression site-associated genes (ESAGs). Between the ESAGs and the VSG gene lies a stretch of conserved AT-rich DNA repeats (the so-called '70-bp repeats') often reaching more than 10 kb (ref. 10). Downstream of these repeats is the VSG gene, which itself is followed by telomeric TTAGGG repeats. A switch in VSG expression can be facilitated by a transcriptional switch from one BES to another, called an in situ switch[11], or by homologous recombination, whereby a previously silent VSG (either from another BES or from elsewhere in the genomic VSG archive) is recombined into the active BES[12].

VSG expression follows a semipredictable and hierarchical pattern, with certain subsets of VSG genes preferentially expressed during specific stages of infection[13,14]. This hierarchy in VSG expression could result from differences in growth rates between cells expressing different VSG isoforms or from differences in the frequency of activation of various

[1]Division of Experimental Parasitology, Faculty of Veterinary Medicine, Ludwig-Maximilians-Universität München, Munich, Germany. [2]Biomedical Center, Division of Physiological Chemistry, Faculty of Medicine, Ludwig-Maximilians-Universität München, Munich, Germany. [3]Institute of Computational Biology, Helmholtz Zentrum München, German Research Center for Environmental Health, Neuherberg, Germany. [4]Department of Molecular Microbiology and Immunology, Johns Hopkins Bloomberg School of Public Health, Baltimore, MD, USA. [5]Present address: Institute for Immunology and Infection Research, School of Biological Sciences, Ashworth Laboratories, University of Edinburgh, Edinburgh, UK. [6]These authors contributed equally: Zhibek Keneskhanova, Kirsty R. McWilliam, Raúl O. Cosentino. ✉e-mail: maria.colome@bmc.med.lmu.de; n.siegel@lmu.de

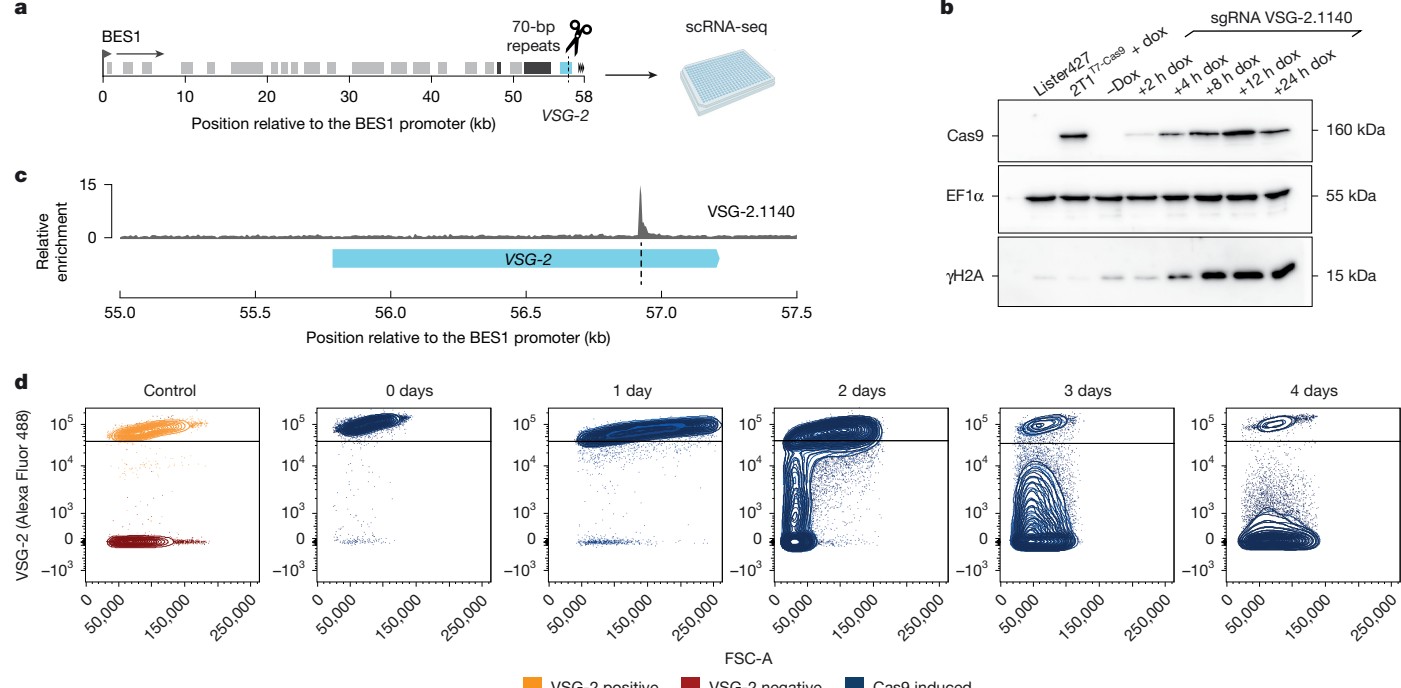

**Fig. 1 | A CRISPR–Cas9 induced DSB in the actively expressed *VSG-2* induces a VSG switch. a**, Schematic map of BES1 with the cut site (dashed line) at nucleotide position 1140 of the *VSG-2* CDS. **b**, Western blot analysis of Cas9 and γH2A protein expression in cells capable of doxycycline (dox)-inducible expression of Cas9 and an sgRNA targeting nucleotide position 1140 of *VSG-2* (sgRNA VSG-2.1140). A wild-type cell line (Lister 427) and a cell line not transfected with the *VSG-2* sgRNA but capable of inducible Cas9 expression (2T1[T7-Cas9]) served as controls. EF1α served as a loading control (*n* = 1). **c**, BLISS coverage around the *VSG-2* CDS after 4 h Cas9 induction in the cell line with sgRNA VSG-2.1140, normalized to the BLISS coverage of wild-type cells.

Shown is the average of two biological replicates. The light blue box represents the *VSG-2* CDS. The on-target DSB position is indicated by a dashed line. **d**, FACS analysis of VSG-2 expression in sgRNA VSG-2.1140 transfected cells in a time course until 4 days post-Cas9 induction. VSG-2 expression was monitored using a fluorophore-conjugated (Alexa Fluor 488) anti-VSG-2 antibody. A minimum of 10,000 events were analysed per sample. On the left panel, VSG-2 positive and VSG-2 negative cell lines are shown as controls (*n* = 1). Graphic in **a** was created using BioRender (https://biorender.com). For gel source data, see Supplementary Fig. 1a.

VSG genes. Both scenarios are supported by experimental evidence and mathematical models[13–19]. However, existing studies of VSG switching have been limited by the inability to determine VSG expression immediately after a switch in individual cells. Typically, VSG expression is assessed at the population level days after the switch event, making it difficult to distinguish differences in growth rates from differences in activation rates. Consequently, it remains unclear whether specific VSG isoforms are preferentially activated and, if so, what mechanism governs this preference during a VSG switch.

To address these questions, we set out to determine the transcriptional profiles of individual cells before, during and after a VSG switch. We developed spliced leader (SL)-Smart-seq3xpress, a highly specific and highly sensitive trypanosome-adapted version of the plate-based Smart-seq3xpress protocol[20]. Using SL-Smart-seq3xpress, we were able not only to determine newly activated VSGs at the single-cell level, but also to determine switching mechanisms, predict sites of DNA recombination and reveal the formation of mosaic VSGs.

Our results indicate that following a double-strand break (DSB) in the active VSG gene, the presence or absence of a suitable homologous repair template, as well as the location of the new VSG gene, determines the DSB repair mechanism and frequency with which a specific VSG gene is activated. Furthermore, our data suggest that *T. brucei* possesses a highly efficient homology search mechanism that identifies suitable genomic regions to repair DSBs in the active VSG.

## DSBs in *VSG-2* trigger VSG switching

Experimentally induced DSBs in the actively transcribed BES have been shown to induce VSG switching with various efficiencies depending on the location of the DSB[19,21]. Thus, to investigate the 'VSG selection mechanism' controlling the hierarchy of VSG expression, we established a Cas9-based system to induce DSBs at various sites across the active BES. We used a cell line containing a tetracycline-inducible SpCas9 and a phage T7 RNA polymerase capable of transcribing single-guide RNA (sgRNA) molecules from a stably transfected plasmid, a similar approach to that used in a recently published study[19].

We designed a sgRNA that directs Cas9 to induce a DSB within the 3′ end of the actively transcribed VSG (*VSG-2*, nucleotide position 1140, Fig. 1a) and tested the inducibility and efficiency of DSB generation. The ability to efficiently generate DSBs in cells transfected with the *VSG-2* sgRNA was confirmed by western blotting and immunofluorescence analysis using an antibody specific to γH2A, a marker of DSBs (Fig. 1b and Extended Data Fig. 1). Breaks labelling in situ and sequencing (BLISS[22]) verified on-target DSB generation on Cas9 induction within the *VSG-2* coding sequence (CDS) (Fig. 1c).

Next, to assess the efficiency by which this approach would lead to a switch in VSG expression, we induced Cas9 expression and analysed VSG-2 expression by flow cytometry using a fluorophore-conjugated VSG-2 antibody. Our results showed that the uninduced starting population was largely VSG-2 positive, but that VSG-2 expression was lost from more than 94% of the population by 4 days post-Cas9 induction (Fig. 1d). Sanger sequencing of complementary DNA (cDNA) isolated from five VSG-2 negative subclones confirmed that the cells had switched expression to a new VSG (Supplementary Data 1). Taken together, these results confirm that CRISPR–Cas9 can be used to reliably and efficiently generate a DSB within the CDS of the actively expressed VSG and to trigger a switch in VSG expression.

## Implementation of SL-Smart-seq3xpress

Bulk RNA-seq cannot distinguish between monogenic and multigenic VSG expression and does not yield information about VSG switch mechanisms nor the site of DNA recombination in individual cells. Therefore, to determine the transcriptome of individual parasites during a VSG switch, we decided to implement an appropriate single-cell RNA sequencing (scRNA-seq) approach.

Following our positive experience with the plate-based Smart-seq2 method[10], we set out to implement a *T. brucei*-adapted version of the most recent Smart-seq protocol: Smart-seq3xpress. Smart-seq3xpress combines full-length transcript detection with a 5′ unique molecular identifier (UMI) counting strategy at a nanolitre scale, significantly reducing the cost per cell without sacrificing sensitivity[20,23]. UMIs are used to eliminate PCR amplification bias and are therefore important for accurate transcript level quantification. Because all mature messenger RNA (mRNA) molecules in *T. brucei* contain a conserved 39-nt SL sequence[24], we omitted the template switch step and instead performed the second-strand cDNA synthesis and amplification using primers annealing to the SL sequence (Fig. 2a). This required us to modify the standard Smart-seq3xpress oligos and therefore the 8-bp UMI, 11-bp UMI-identifying tag and partial Tn5 motif were moved from the template switch oligo to the oligo(dT) primer (Fig. 2a and Extended Data Fig. 2a). To account for *T. brucei* cells containing significantly less RNA than the mammalian human embryonic kidney 293FT cells with which the Smart-seq3xpress protocol was developed, we adjusted the transposase and oligo(dT) primer concentrations to maximize transcript diversity and the number of UMI-containing reads without losing the transcript internal reads that do not contain UMIs but are important for distinguishing VSG isoforms (Extended Data Fig. 2b,c).

Next, we assessed the specificity of our sequencing approach. Following the introduction of patterned flow cells in the latest generation of Illumina sequencers, several groups have reported library index hopping as a significant source of error[25]. Such index hopping can result in the incorrect assignment of reads from one cell to another and would limit our ability to investigate the regulation of mutually exclusively expressed genes, such as the VSG (Extended Data Fig. 2d). To address this challenge, we implemented scSwitchFilter (https://github.com/colomemaria/scSwitchFilter), a bioinformatic tool that removes index-hopped reads based on the assumption that reads with the correct gene–UMI–index combination should be more abundant than reads with the same gene–UMI combination arising from an index hopping event (Extended Data Fig. 2d). Application of this filtering approach drastically decreased the observed number of index hopping events, with the median UMI count from a single cell now 294-fold higher than that from control wells not containing a cell (Extended Data Fig. 2e). We therefore applied the index hopping filtering pipeline to all quantitative transcriptome analyses.

## Single-cell VSG transcript measurements

Having optimized our SL-Smart-seq3xpress pipeline, we compared its sensitivity (the number of genes and UMIs detected per cell) to the best available 3′ Chromium Single Cell (10X Genomics) dataset for *T. brucei*[26]. At the same sequencing depth, we found SL-Smart-seq3xpress to be significantly more sensitive, detecting a median of 2,876 genes and 4,640 UMIs per cell versus 1,052 genes and 1,552 UMIs detected in the published 10X Genomics dataset (Fig. 2b and Extended Data Fig. 2f, respectively). At a higher sequencing depth (1 million reads per cell sequenced) we were able to detect a median of 16,797 transcripts per cell (Fig. 2c). This corresponds to roughly 84% of the predicted total number of mRNA molecules in a single cell[27]. Thus, our SL-Smart-seq3xpress approach ranks among the most sensitive scRNA-seq methods reported[28].

To test whether SL-Smart-seq3xpress was able to both reliably detect, and distinguish between, VSG transcripts originating from different cells, we mixed isogenic cells expressing one of two different VSGs, either *VSG-2* or *VSG-13* (refs. 29,30), in equal proportions and generated sequencing libraries. For comparison, we generated another sequencing library from the same mixture of cells using the 5′ Chromium pipeline. We defined a cell's VSG expression as 'mutually exclusive' when more than 80% of the cell's *VSG-2* and *VSG-13* transcripts mapped to only one of the two VSG genes. Using our SL-Smart-seq3xpress pipeline, we observed clean separation between *VSG-2* and *VSG-13* expressers, with 100% of the cells reaching the defined threshold for mutually exclusive expression (Fig. 2d). By contrast, when cells from the same mixed population were sequenced using the 5′ Chromium method, 11% of the cells did not surpass the defined threshold (Fig. 2d). Similar observations were made when *T. brucei* and *L. mexicana* cells were mixed during a 3′ Chromium analysis[26]. When we increased the threshold from 80 to 90%, we still found that 100% of the cells sequenced using SL-Smart-seq3xpress show unambiguous mutually exclusive VSG expression (Fig. 2d). For the 5′ Chromium platform, the percentage of cells not meeting the threshold increased from 11 to 14%, similar to that reported in a recent VSG analysis using the platform[31]. Our low background is consistent with previous observations suggesting that plate-based scRNA-seq approaches suffer less from ambient RNA contamination than droplet-based approaches[32]. Overall, our SL-Smart-seq3xpress pipeline, combined with the bioinformatic clean-up of index-hopped reads, seems to be very well suited to the analysis of mutually exclusively expressed genes such as VSGs.

## *VSG-2* cuts activate only telomeric VSGs

We now applied our approach to investigate VSG selection mechanism by monitoring VSG expression in single cells before and at various times after induction of a DSB within *VSG-2*. *VSG-2* is the active VSG gene in most laboratory-adapted isolates. We generated a DSB at nucleotide position 1140 (Fig. 3a) of the actively expressed *VSG-2* and prepared SL-Smart-seq3xpress sequencing libraries from 369 cells at 0, 1, 2, 3, 4 and 10 days after Cas9 induction. This was done for two biological replicates for a total of 738 cells per time point. The scRNA-seq data indicated that the generation of a DSB within *VSG-2* led to a rapid decrease in the proportion of cells that dominantly expressed *VSG-2* (Fig. 3b). In these cells the total number of VSG transcripts dropped by almost tenfold, pointing to a transcriptional arrest of the active BES, while no other VSG transcripts were upregulated (Extended Data Fig. 3a–d). By 2 days post-Cas9 induction these cells comprised most of the population. However, we then began to observe that a broad but well-defined set of VSG genes had been activated at the population level, including VSGs located in BESs, subtelomeric VSG arrays, minichromosomes and metacyclic expression sites (MES, expression sites that are active in the infectious metacyclic form before BES activation) (Fig. 3b,c). The similarity of the set of activated VSGs between the two replicates was striking, suggesting hard-wired mechanisms leading to the preferential activation of some VSGs over others, as observed previously[19,33]. By day 10 postinduction, most cells expressed a single dominant VSG, and several switched clones, particularly those expressing VSG-9 and VSG-18, had begun to outgrow, suggesting VSG-specific fitness advantages contribute to a reduction in VSG expression heterogeneity within the population (Fig. 3b). Given that these VSGs are among the largest in the genome, this observation suggests that, in vitro, growth dynamics are not primarily governed by VSG length[18].

Significantly, when we considered all newly activated VSGs, a consistent pattern emerged: every newly activated VSG was situated immediately adjacent to a telomere. Out of the activated VSGs, 12 were located in a BES, 5 in a metacyclic expression site, 19 were located on minichromosomes and 2 were the telomere-proximal

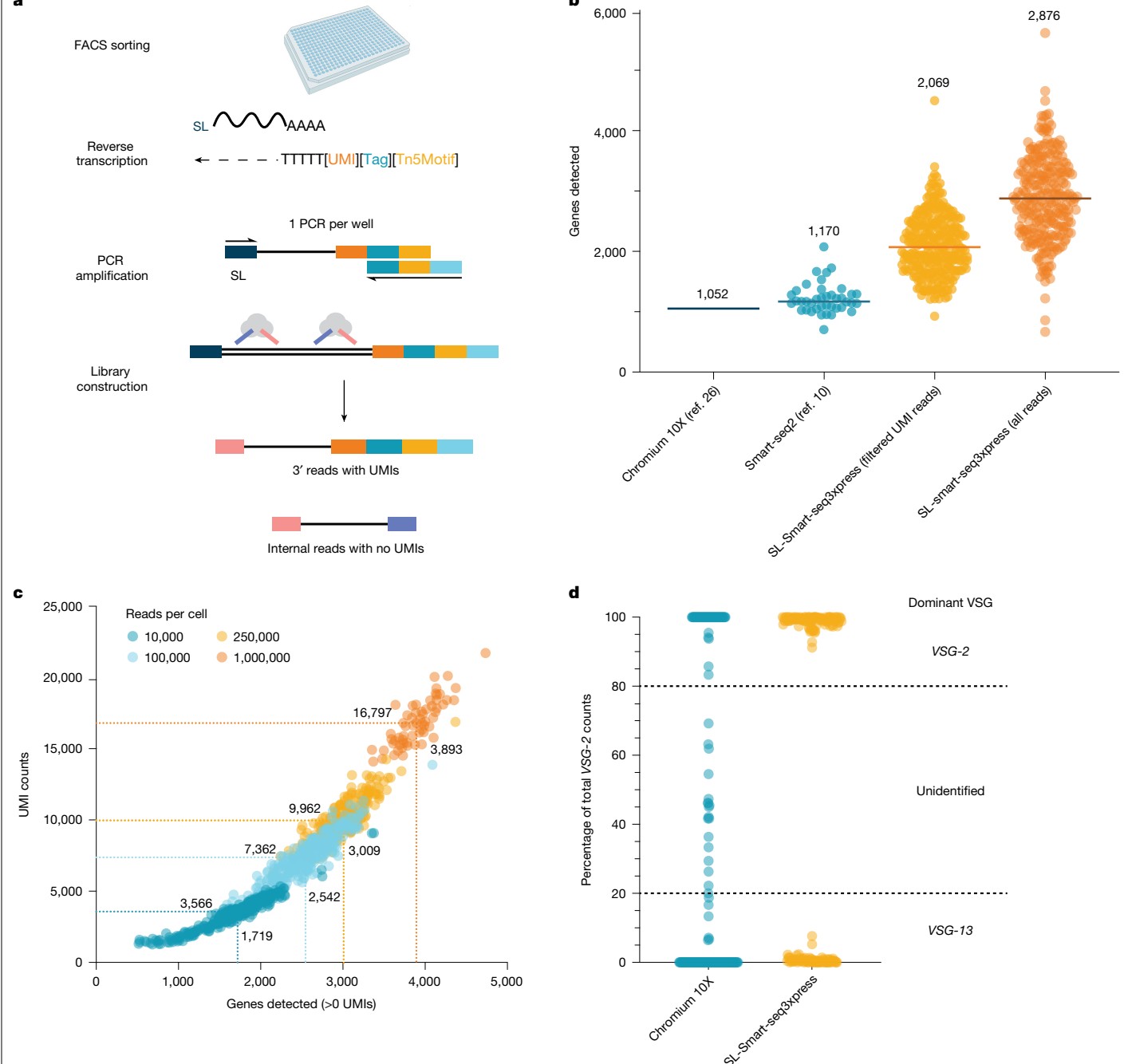

**Fig. 2 | SL-Smart-seq3xpress is a highly sensitive and specific scRNA-seq approach. a**, Schematic of SL-Smart-seq3xpress library preparation. **b**, Comparison of the median number of genes detected by Chromium 10X (data from Briggs et al.[26]), Smart-seq2 (data from Müller et al.[10]) or SL-Smart-seq3xpress. The median UMI–gene count is shown as a number above a dataset and as a bold line. For the Smart-seq2 and SL-Smart-seq3xpress datasets, each dot represents an individual cell. For the Chromium 10X dataset, the line represents the median UMI–gene count. Libraries are downsampled to 75,000 reads per cell. Number of cells analysed: Chromium 10X, 8,599; Smart-seq2, 40; SL-Smart-seq3xpress, 292. **c**, Detected UMIs versus genes in a SL-Smart-seq3xpress library at increasing read depth. Each single cell is represented by a coloured dot. The median gene versus UMI count for each read depth is represented by the dotted lines and numbers. **d**, Percentage of *VSG-2* transcript counts, relative to the sum of *VSG-2* and *VSG-13* transcript counts, in Chromium 10X and SL-Smart-seq3xpress single-cell libraries prepared from mixed populations of *VSG-2* expressing (P10 cell line) and *VSG-13* expressing (N50 cell line) cells. The thresholds for defining a cell as *VSG-2* or *VSG-13* expressing (above 80% and below 20%, respectively) are indicated by the dotted lines. Total number of cells analysed: Chromium 10X, 185; SL-Smart-seq3xpress, 185. Graphic in **a** was created using BioRender (https://biorender.com).

VSG gene in a subtelomeric VSG array (Supplementary Table 1). To confirm that this observation was not unique to the chosen cut site, we generated two additional cell lines in which Cas9 generated a DSB either at another site in the *VSG-2* CDS or between the second set of 70-bp repeats on BES1 and the *VSG-2* CDS (Extended Data Fig. 4a). As before, inducible DSB generation was verified by western blotting using γH2A as a marker and on-target DSB generation was validated by BLISS (Extended Data Fig. 4b,c). Following the generation of a DSB at either of these cut sites, we observed a pattern of VSG activation that was similar to that of the original cut site (Extended Data Fig. 4d–f). In all cases, cuts within *VSG-2*, or just upstream, led to the activation of a telomere-adjacent VSG.

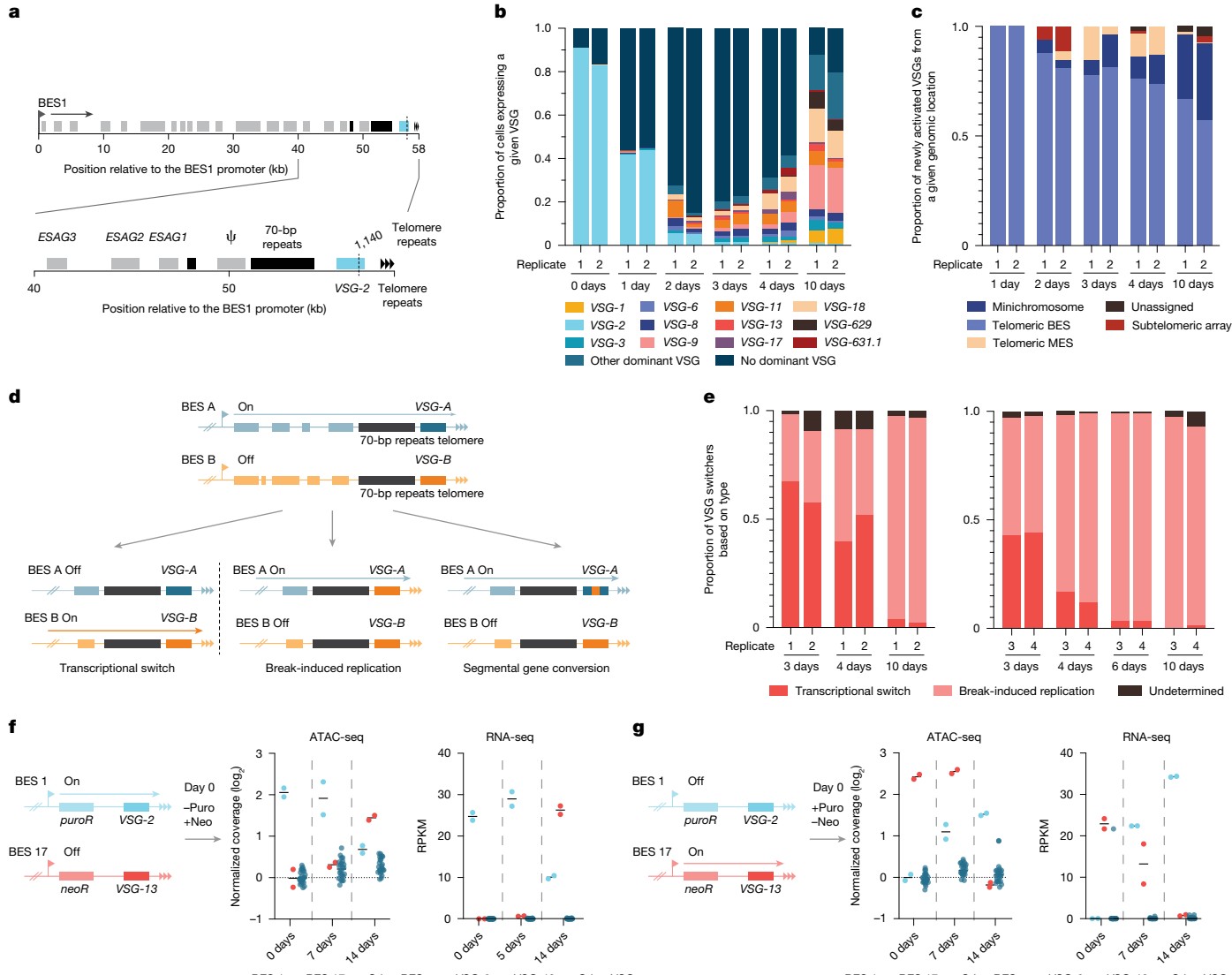

**Fig. 3 | DSBs in the active *VSG-2* CDS led to activation of telomere-adjacent VSGs. a**, Schematic map of BES1 indicating cut sites (dashed lines). **b**, VSG expression in single cells before and after the induction of a DSB in the *VSG-2* CDS at nucleotide position 1140, as measured by SL-Smart-seq3xpress, two biological replicates (R). The total number of cells analysed per time point and replicate is as follows: 0 days R1, 312; 0 days R2, 302; 1 day R1, 308; 1 day R2, 283; 2 days R1, 147; 2 days R2, 271; 3 days R1, 305; 3 days R2, 255; 4 days R1, 289; 4 days R2, 286; 10 days R1, 336; 10 days R2, 323. **c**, Proportion of cells at each time point after DSB induction from **b** expressing a new VSG from a given genomic location. 'Unassigned' refers to newly activated VSGs for which the original location is not known. **d**, Diagram illustrating the different types of VSG switching mechanism. **e**, Left, percentage of cells at days 3, 4 and 10 post-Cas9 induction

in each biological replicate that switched VSG expression by a given mechanism. Right, same as the left for two more biological replicates with more intermediate time points. **f**, Time course of transcriptional switcher selection from BES1 (*VSG-2*) to BES17 (*VSG-13*) by the addition of neomycin (+neo) and removal of puromycin (−puro) drug selection. ATAC-seq coverage on BES1, BES17 and all other BESs (middle), and transcript reads per kilobase per million mapped reads (RPKM) for their respective VSGs (right). 'Other VSGs' refers to VSGs located in BESs other than BES1 and BES17. Horizontal lines represent the mean of two biological replicates. **g**, Time course of transcriptional switcher selection as in **f** but from BES17 (*VSG-13*) to BES1 (*VSG-2*) by the addition of puromycin (+puro) and removal of neomycin (−neo) drug selection.

## BES1 activation confers growth advantage

To reveal why only telomere-adjacent VSGs are activated, we sought to determine the mechanism (transcriptional switch, break-induced replication (BIR) or segmental gene conversion) by which each VSG was activated (Fig. 3d).

The high sensitivity of SL-Smart-seq3xpress allowed us to detect transcripts from ESAGs and thus determine the site of recombination within BES1 or if a new BES had been activated (Extended Data Fig. 5a). Our data indicated that most recombination events occurred in (or near) the 70-bp repeats upstream of *VSG-2* (Extended Data Fig. 5b). However, recombination was not restricted to the 70-bp repeats; in particular, for

switchers to *VSG-11*, *VSG-9*, *VSG-18* and *VSG-8*, we observed a considerable fraction of cells whose transcriptional profiles suggest recombination between ESAGs (Extended Data Fig. 5b,c). In addition, we observed a much higher percentage of transcriptional switchers at early time points than at later time points. In our initial switching time course analysis (*n* = 2), we observed that the percentage of transcriptional switchers decreased from 62.5% at day 3 to 46.5% at day 4 and 3% at day 10 (Fig. 3e, left). In an additional switching time course analysis (*n* = 2), we observed a similarly rapid decrease in the percentage of transcriptional switchers (Fig. 3e, right). By day 6, 95.4% of cells were expressing a VSG recombined into BES1 through BIR (Fig. 3e, right). When we cut at the second *VSG-2* cut site (nucleotide 782), we again observed a loss of

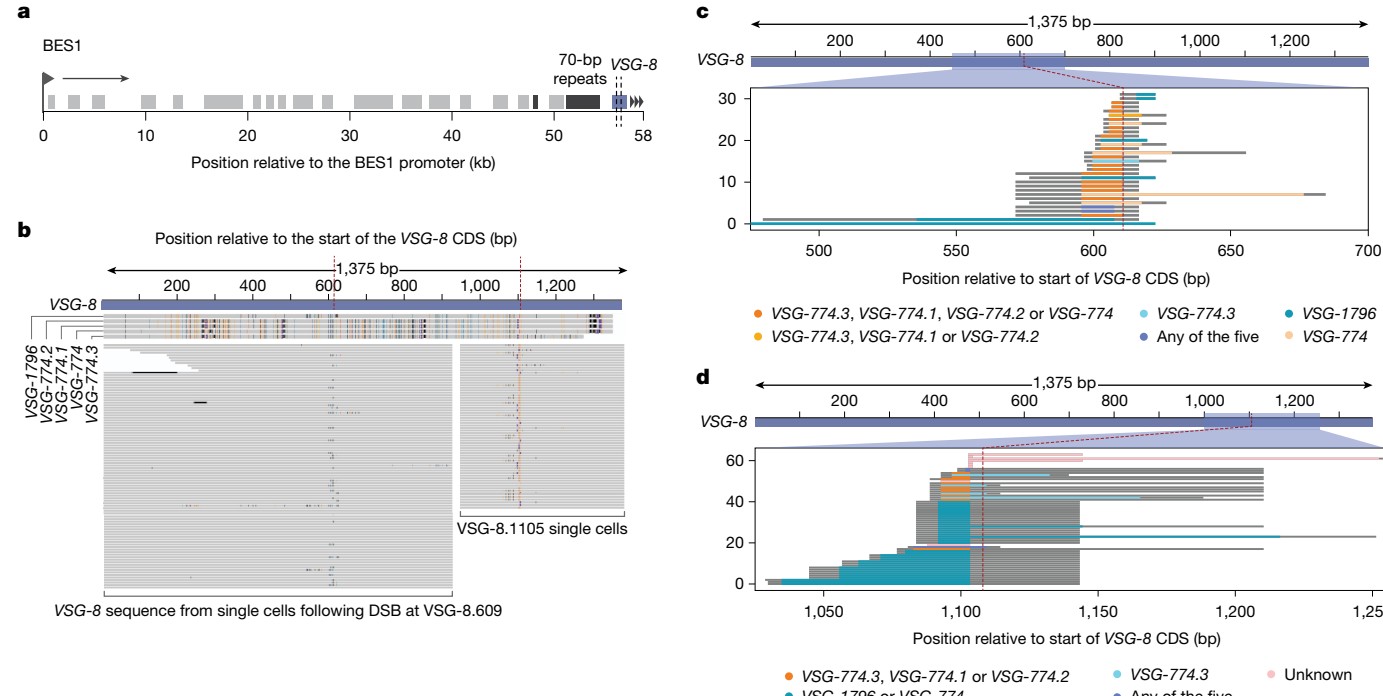

**Fig. 4 | DSB in *VSG-8* CDS leads to switching by segmental gene conversion.**
**a**, Schematic map of the recombinant BES1 (expressing *VSG-8*) with the cut sites (dashed lines) on the *VSG-8* CDS. **b**, De novo assembled VSG transcripts from individual cells after 4 days of DSB induction at nucleotide positions 609 and 1105 of the *VSG-8* CDS, aligned against the *VSG-8* sequence. Five homologous VSG genes or pseudogenes (potential 'donors') found in the *T. brucei* genome by BLAST search are also included on top of the alignment. Mismatched bases are coloured based on the nucleotide identity. **c**,**d**, Recombinant fragments based on de novo assembled VSG transcripts for individual cells after 4 days of DSB induction at nucleotide positions 609 (**c**) and 1105 (**d**) of the *VSG-8* CDS. The potential 'donor' VSG and the integrated stretch length are depicted by the coloured lines. Grey extensions on the lines represent the maximal sequence length that could have been recombined (that is, stretch until the next nucleotide difference between *VSG-8* and the possible donor(s)). All single-cell data are derived from two biological replicates per cut site.

transcriptional switchers across time, with cells that had predominantly switched by recombination around the 70-bp repeats dominating the population at day 10, confirming that our results were not unique to the nucleotide 1140 cut site (Extended Data Fig. 5d,e). This observation suggests that transcriptional switchers that had activated another BES were either outcompeted by cells that had maintained BES1 expression, or had switched back to transcribing BES1 once the DSB was repaired. Both of these scenarios suggest that, at least in vitro, there is a strong fitness advantage associated with BES1 transcription.

Previously, it has been suggested that following a transcriptional switch, the previously active BES remains in an open state, possibly 'poised' for reactivation[34]. To test this possibility, we used assay for transposase-accessible chromatin sequencing (ATAC-seq) to determine whether the chromatin landscape of BES1 remained open, and thus poised for re-expression, following a transcriptional switch to another BES. To select for transcriptional switchers, we used a cell line containing a puromycin drug resistance gene in BES1 and a neomycin resistance gene in BES17 (ref. 35) (Fig. 3f). By treating with either drug, we could induce transcriptional switching between BES1 and BES17 and select for cells expressing either *VSG-2* from BES1 or *VSG-13* from BES17.

ATAC-seq assays performed 0, 7 and 14 days after replacing puromycin with neomycin to activate BES17 with *VSG-13*, indicated that even at day 14 BES1 was more open than the silent BESs (Fig. 3f). We also observed that *VSG-2* transcript levels were only partially reduced (Fig. 3f). As the experiment was performed in bulk, we cannot say for certain that BES1 stayed open and 'poised' for reactivation, whereas BES1 may have stayed poised in some cells; in other cells BES1 may have never stopped being transcribed.

However, when performing the reverse experiment (replacing neomycin with puromycin to activate BES1 with *VSG-2*), we found that *VSG-13* transcript levels decreased rapidly (Fig. 3g). In addition, ATAC-seq data indicated that BES17 was closed by day 14. Thus, although these assays cannot rule out or confirm that a BES stays poised after a transcriptional switch, they indicate that a switch away from BES1 follows different dynamics from a switch from BES 17 to BES1, again suggesting that there may be an advantage to expressing BES1. In our *VSG-2* cutting experiments, cells that switched BES expression may therefore be outcompeted from the population by BES1 expressors, or represent an intermediate stage transcribing an alternate BES to satisfy VSG requirements until the DSB is repaired in BES1. Following successful repair, the cells may then re-express the poised BES1.

## Impact of repair template availability

The question that remained was why would the parasite have a reservoir of more than 2,500 VSG genes and only activate the telomere-adjacent VSGs by transcriptional or BIR-mediated switches? In our *VSG-2* cutting experiments, we never observed activation of a new VSG gene by segmental gene conversion. Given the importance of local sequence homology in DNA repair by gene conversion, this led us to analyse the *VSG-2* coding sequence. We saw that *VSG-2* is unusual among VSG genes, in that its CDS has very little sequence similarity to any other VSG gene in the genome. Proposing that the 'uniqueness' of the *VSG-2* CDS could influence the VSG selection mechanism, we determined whether the outcome of our switching experiments would change if we cut in an active VSG that had a high degree of sequence similarity to other VSGs.

To generate cell lines expressing an active VSG gene that shared sequence homology with at least one other VSG gene or pseudogene

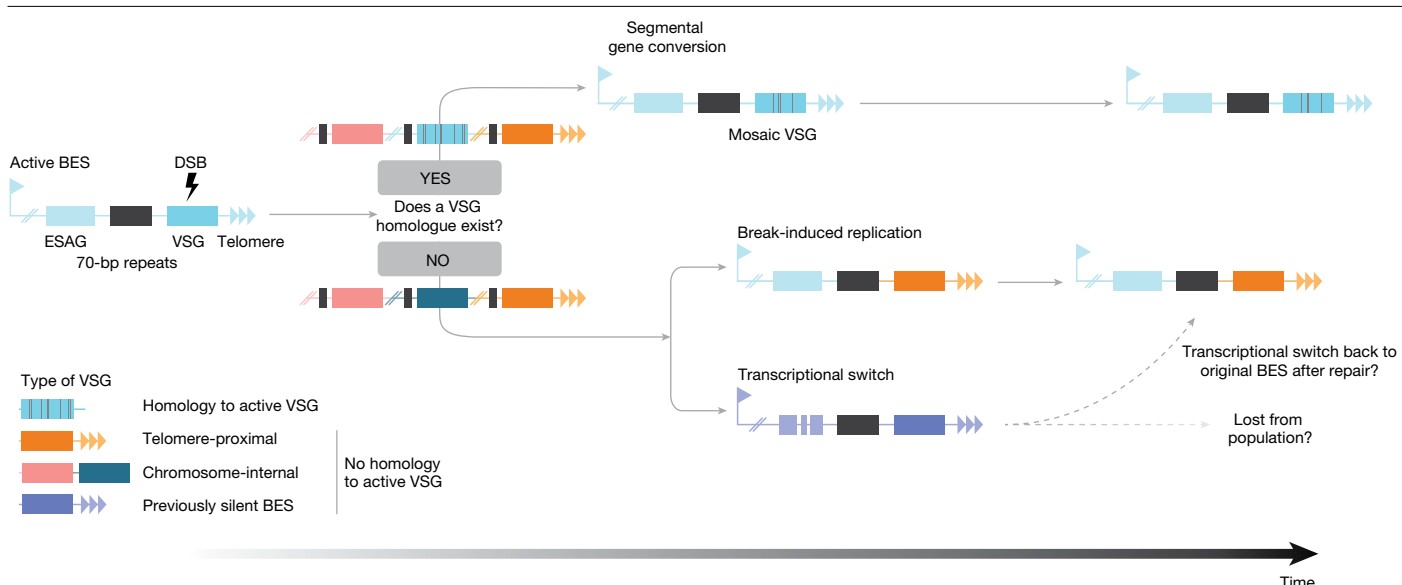

**Fig. 5 | Model of VSG selection mechanism.** Diagram summarizing how a DSB in the VSG CDS seems to be repaired in the presence or absence of a suitable repair template.

in the genome, we induced a DSB at position 1140 of *VSG-2*, subcloned surviving switchers, determined the actively expressed VSG by Sanger sequencing of the clones' cDNA, and selected a clone expressing *VSG-8* from BES1 (Extended Data Fig. 6a). *VSG-8* has a second copy in the genome and shares regions of DNA sequence homology with at least five other VSG genes and pseudogenes in the genome, as determined by BLAST.

In this *VSG-8* expressing clone, we then induced a DSB in the *VSG-8* CDS (either at nucleotide position 609 or 1105, Fig. 4a,b) using sgRNAs that were designed to target exclusively *VSG-8*, and not one of the similar VSG genes. Efficient DSB induction was confirmed by analysis of γH2A signal intensity using western blotting (Extended Data Fig. 6b) and by BLISS (Extended Data Fig. 6c). We observed a milder restriction of population growth upon DSB generation at both sites, compared to that observed following the cuts in *VSG-2* (Extended Data Fig. 6d–f). SL-Smart-seq3xpress libraries were prepared from cells before and 4 days after Cas9 induction for each of the two cut sites for two biological replicates each.

Analysing the sequencing reads, we were surprised to find that there was no evidence of DNA recombination within the 70-bp repeats nor of transcriptional switching in more than 99.4% cells induced to have a DSB in the *VSG-8* CDS (Extended Data Fig. 6f). However, the reads mapping onto the actively expressed VSG at 4 days post-Cas9 induction presented a few single-nucleotide polymorphisms (SNPs) around the expected DSB site when comparing them to the *VSG-8* sequence expressed in uninduced cells. Using the scRNA-seq reads, we generated de novo VSG assemblies for the individual cells and found that the location, identity and number of SNPs varied between the sequenced cells, suggesting that the observed repair had occurred in a different manner in each cell (Fig. 4b). It is important to note that we never observed SNPs around the cut site when we induced a DSB in the *VSG-2* CDS. Because non-homologous end-joining does not occur in *T. brucei*[36] and we observed no evidence of DNA resection and homologous recombination upstream of the break site, we investigated whether segmental gene conversion, using one of the (pseudo)genes similar to *VSG-8* as a template, was responsible for the observed generation of sequence variation at the active VSG gene.

Using BLAST, we searched for sequences similar to that of *VSG-8* and aligned them to the de novo assembled VSG transcripts that we had generated for each single cell. We observed that most of the novel

sequence stretches around the DSB position had an exact match in at least one of the VSG genes similar to *VSG-8* present in the genome. The many sequence combinations observed suggests that (1) the same VSG gene was not always used as the repair template and that (2) the transferred DNA segment was of variable length (Fig. 4c,d). The VSGs that contained the matching SNPs were all pseudogenes and were located in different subtelomeric arrays within the genome. Thus, our data suggest that in the presence of a suitable repair template (VSG gene or pseudogene), DSBs in the active VSG gene are preferentially repaired by segmental gene conversion, leading to the generation of new 'mosaic' VSG genes.

To confirm that our observed switching phenotype was not unique to *VSG-8* expressing cells, we selected another switched clone expressing *VSG-11* in BES1 and designed sgRNAs to induce DSBs at nucleotide positions 519 or 729 of the *VSG-11* CDS (Extended Data Fig. 6h). *VSG-11* also has a second copy in the genome and shares regions of DNA sequence homology with at least one other VSG. Sanger sequencing of VSG cDNA amplified from cells induced to generate breaks at either site in *VSG-11* confirmed that mosaic VSGs had again been generated by segmental gene conversion (Extended Data Fig. 6i).

The observation that almost every cell expressing *VSG-8* or *VSG-11* repaired its DSB in the VSG CDS by segmental gene conversion, rather than by BIR as we observed for DSBs in *VSG-2*, suggests that *T. brucei* first searches for homologous VSG sequences to use as DNA repair templates before starting to resect the BES towards the 70-bp repeats.

## Discussion

To investigate the mechanism of VSG selection, we leveraged CRISPR–Cas9 technology to create targeted DSBs at specific sites along BES1 and developed a highly sensitive, trypanosome-tailored scRNA-seq protocol. Our findings indicate that the VSG selection mechanism is influenced by the presence (or absence) of DNA homology regions for repair.

From these results, we propose a model with two potential outcomes for a DSB in an active VSG gene: (1) if a homologous region exists in the genome, the DSB is repaired through segmental gene conversion, involving crossovers up- and downstream of the break site. This process can result in mosaic VSGs. (2) In the absence of a

homologous region, the DSB is repaired by means of BIR. This leads to the duplication and activation of telomere-adjacent VSG genes into the active BES (Fig. 5).

This model is supported by our observations following DSBs in actively transcribed *VSG-8* or *VSG-11* genes, for which homologous regions are available in different genomic loci. In these cases, segments from pseudogenes in subtelomeric VSG arrays were incorporated into the active VSG, forming mosaic VSGs. No VSGs from other BESs or minichromosomes were copied by means of BIR, suggesting that *T. brucei* possesses an effective homology search mechanism and prefers repair by segmental gene conversion over BIR at the 70-bp repeats.

Our data also showed that DSBs in the actively transcribed *VSG-2*, which is atypical among VSGs in that it shares very little homology with other VSG genes, were repaired by BIR. Immediately after DSB induction, we observed a strong decrease in total VSG transcript levels and in BES1 ESAG transcripts, probably caused by a break-induced transcriptional arrest of the BES while the homology search was continuing. At the same time, we observed a high proportion (40–60%) of transcriptional switchers. As we detected almost no transcriptional switchers at later time points (less than 3% at day 10), we speculate that they are either lost from the population due to a fitness disadvantage or that they represent an intermediate stage until the cells can switch back to BES1 transcription once the DSB has been repaired. Either scenario would suggest that in Lister 427 isolates, transcription of BES1 confers a strong fitness advantage in vitro over transcription of other BESs. This apparent selective pressure for BES1 may explain the low number of VSG switchers observed in this isolate compared to other isolates.

Combining CRISPR–Cas9 technology to induce cuts in both *VSG-2* and non-*VSG-2* expressing cell lines with transcriptional profiling of single cells, we were able to reveal the critical role of available homologous DNA sequence in determining the outcome of DSB repair. Furthermore, our findings suggest that mosaic formation might be the 'preferred' mechanism for antigen activation following a DSB in the active VSG. It is fast, does not involve resection and keeps the original BES active. Mosaic VSG formation is proposed to emerge as natural infections progress, serving as a mechanism for diversifying the range of antigenically distinct VSG isoforms[14,37]. Although the exact mechanisms driving mosaic formation are still not completely understood, our results indicate that DNA homology plays a fundamental role in this process. It has been suggested that mosaic formation could occur either shortly before the functional VSG is recombined into the active BES, or within the active BES itself at the time of the VSG switch[14]. Our findings indicate the latter may be the case. Existing γH2A chromatin immunoprecipitation with sequencing and BLISS data indicate that DSBs naturally occur within the active VSG gene CDS[38,39]. Therefore, it seems plausible that mosaic formation occurs more frequently in vitro than previously recognized, particularly given the extensive research conducted on the atypical VSG isoform, VSG-2.

In summary, our work sheds light on the decades-old question of why some VSGs are activated more frequently than others. We found that the availability of homologous DNA for DSB repair and the genomic location of VSGs are key factors in determining the hierarchy of VSG activation. Our data also demonstrate that tailored, highly sensitive scRNA-seq approaches not only facilitate the study of cell-to-cell heterogeneity at the level of the transcriptome but to also discover underlying genomic changes. We believe that the ability to link genomic and transcriptional diversity of pathogens at the single-cell level will be a powerful tool to dissect the evolution of drug resistance and to aid in the design of more robust drugs.

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

# Methods

## Trypanosome culture and genetic manipulation

Bloodstream form *T. brucei* Lister 427, P10 and N50 (ref. 35), 2T1[T7] (ref. 40) and 2T1[T7 Cas9] trypanosomes along with their derivatives, were maintained at 37 °C and 5% $CO_2$ in HMI-11 supplemented with 10% fetal calf serum and the appropriate selective drug[41]. Cells were maintained below $2.0 \times 10^6$ cells per ml. Cells were electroporated as previously described[42]. The 2T1[T7 Cas9] cell line was generated by transfecting 2T1[T7] cells with the pRPa[Cas9] plasmid[43], which integrates into a tagged ribosomal DNA (rDNA) spacer region[40]. Cas9 expression was induced with $1 \, \mu g \, ml^{-1}$ doxycycline. For the SL-Smart-seq3xpress switching assays, Cas9 induction was maintained throughout the time course. To generate clonal populations of VSG switched cells following the induction of a DSB, the induced population was diluted to 8–10 cells per ml in HMI-11 and spread across the wells of a 96-well plate in 100-μl volumes. Plates were left for 5 days for the cells to recover. Cells were counted with a Beckmann Coulter cell counter.

## Sanger sequencing of expressed VSG transcripts

RNA was extracted from $5.0 \times 10^6$ cells using the NucleoSpin RNA Kit (Macherey & Nagel) according to the manufacturer's instructions. RNA was stored at −80 °C. First strand cDNA was synthesized from 5 μg of the extracted RNA using the SuperScript II Reverse Transcriptase (Invitrogen) and the oligo(dT)$_{12-18}$ primer as per the manufacturer's instructions. First strand cDNA was stored at −20 °C. Expressed VSG transcripts were amplified from the first strand cDNA by PCR using a forward primer specific to the SL sequence (5′-GACTAGTTTCTGTACTAT-3′) and a reverse primer specific to the conserved 3′ sequence of VSG mRNAs (5′-CCGGGTACCGTGTTAAAATATATC-3′). For each PCR reaction, 1 μl of 1:50 diluted first strand cDNA was used. PCR products were visualized following agarose gel electrophoresis and the VSG amplicon purified from the agarose using the Nucleospin Gel and PCR Clean-up Kit (Macherey & Nagel) as per the manufacturer's instructions. Sanger sequencing of at least 100 ng of purified PCR product was performed by Eurofins Genomics using either of the primers used in the VSG amplification PCR.

## sgRNA design and cloning

sgRNA target sequences were designed with Protospacer Workbench[44] (v.0.1.0 beta) using the Lister 427 2018 genome assembly (ref. 10, https://tritrypdb.org/) as the reference database and optimized for use with SpCas9. sgRNA target sequences were selected according to their Bowtie Score (a measure of off-target Cas9 activity) and the Doench-Root-Activity score (a measure of sgRNA activity). As described in ref. 43, 'aggg' was added to the forward primer sequence and 'caaa' to the reverse primer sequence to create the BbsI cloning sites. The target sequences were cloned into the pT7[sgRNA] plasmid and transfected into the 2T1[T7 Cas9] cell line as previously described in ref. 43. The pT7[sgRNA] plasmid integrates into a random rDNA spacer region.

## Western blot analysis of Cas9 and γH2A expression

Total protein extract from $2.0 \times 10^6$ cells was boiled in 1× lysis buffer (1:3 4× Laemmli:1× RIPA, 2 mM dithiothreitol (DTT), 1% β-mercaptoethanol) and separated on a 12.5% SDS−PAGE gel. Separated proteins were transferred onto a methanol-equilibrated PVDF (polyvinyl difluoride) membrane using a Bio-Rad Mini Trans Blot Cell according to the manufacturers' instructions. To visualize transferred proteins, the membrane was stained with 0.5% Amido black solution (in 10% acetic acid). Destaining was performed with 1× destaining solution (25% isopropanol, 10% acetic acid). The blotted PVDF membrane was cut into three according to the prestained protein ladder before blocking: above 70 kDa for the detection of SpCas9, between 70 and 25 kDa for the detection of the EF1α loading control and below 25 kDa for the detection of γH2A.

Cas9 and EF1-α blots were blocked in 5% milk/PBS-T and γH2A blots were blocked in 3% BSA/PBS-T at room temperature. The membranes were washed three times with 1× PBS-T and primary antibody incubation was performed overnight at 4 °C. The primary antibodies were used at the following dilutions: anti-Cas9 (1:1,000 in 5% milk/PBS-T, Active Motif, clone 7A9-3A3); anti- EF1α (1:20,000 in 1% milk/PBS-T, EMD Millipore Corporation, clone CBP-KK1) and anti-γH2A (1:200 in 1% milk/PBS-T, from L. Glover, Institut Pasteur). After washing the membranes three more times with PBS-T, the following horseradish peroxidase-conjugated secondary antibodies were used: for Cas9, anti-mouse (1:10,000 in 1% milk PBS-T, GE Healthcare, code NA931V); for γH2A, anti-rabbit (1:2,000 in 1% milk/PBS-T, GE Healthcare, code NA934V) and for EF1-α, anti-mouse (1:10,000 in 1% milk/PBS-T, GE Healthcare, code NA931V). Following secondary incubation, the membrane was washed three times with PBS-T and once more with PBS. For signal detection, the Immobilon Western chemiluminescent horseradish peroxidase substrate was used according to the manufacturers' instructions. The signal was visualized on a ChemiDoc MP Imaging System (v.3.0.1.14).

## Immunofluorescence analysis of γH2A expression

Immunofluorescence analysis of γH2A expression was performed as previously reported[45]. At least 250 cells were analysed per sample. Images were acquired with a Leica DMi8 inverted fluorescence microscope with the Leica Application Suite X (LAS X) software (v.3.7.6) and processed with Fiji (v.2.0).

## FACS analysis of VSG expression

Fluorescence-activated cell sorting (FACS) analysis of VSG-2 expression was performed on live cells and therefore all steps were performed at 4 °C to prevent internalization of the VSG-2 antibody. Cells were stained immediately before analysis. For each replicate, $1.0 \times 10^6$ cells were collected by centrifugation and incubated with fluorescently conjugated anti-VSG-2 (ref. 46) diluted 1:500 in HMI-11 in the dark. Cells were washed three times with 1× TDB and resuspended in 400 μl of 1× TDB. Cells were stained with $1 \, \mu g \, ml^{-1}$ propidium iodide for the identification of dead cells. Samples were processed on a FACS Canto (BD Biosciences) and 10,000 events were captured per sample. Data were processed using FCS Express software (v.7). Gates were applied to remove cellular debris (FSC-A versus SSC-A) and remove doublets (FSC-A versus FSC-H).

## Single-cell sorting for SL-Smart-seq3xpress library preparation

Single cells were sorted into 384-well plates for SL-Smart-seq3xpress library preparation by flow cytometry using a FACS Fusion II cell sorter (BD Biosciences) and a 100-μm nozzle within a safety cabinet. The sorter was calibrated according to the manufacturer's protocol before collecting cells to reduce the time cells were held before sorting, thereby reducing cell death. A 384-well plate adaptor was installed and prechilled to 4 °C. Correct droplet positioning within wells was verified visually by sorting empty droplets onto a covered 384-well plate before every plate was sorted. Next, $5.0 \times 10^6$ cells were collected by centrifugation at 4 °C and washed twice in sterile filtered ice-cold 1× TDB. The cells were resuspended in 1 ml of ice-cold filtered 1× TDB, stained with $1 \, \mu g \, ml^{-1}$ propidium iodide and brought immediately to the sorter on ice. Populations were gated to remove cellular debris, doublets and dead cells as described above. As a consequence of our tight gating strategy (Extended Data Fig. 7), we probably enriched for cells in G1 and excluded larger cells in G2. Plates prepared with lysis buffer were thawed individually immediately before sorting and placed within the precooled adaptor. Single cells were sorted using the 'single cell' purity option into the appropriate wells, and the plates were immediately sealed with an aluminium foil and moved to dry ice before longer term storage at −80 °C. Sorted plates were not stored for more than 1 month before library preparation.

## Generation of SL-Smart-seq3xpress RNA spike-ins

From the standard ERCC RNA spike-in set of 92 sequences, ten with a size of around 500 nt and roughly 50% G+C content were selected and synthetic spike-in DNA fragments were ordered from IDT, adding a homology region for cloning, the T7 promoter sequence and the SL sequence on the 5′ end and a homology region for cloning on the 3′ end. The fragments were cloned into a pBSIIKS+ plasmid digested with SacI and BamHI (NEB) using Infusion (Takara) and transformed into Stellar cells. Plasmids were then extracted, linearized with BamHI and in vitro transcription and polyadenylation was performed using HiScribe T7 ARCA mRNA Kit (with tailing) from NEB, following the recommended procedure. The obtained RNA from each of the ten spike-in sequences were mixed and aliquot dilutions of the spike-in mix were generated and stored at −80 °C until usage. Annotations based on the spike-in fasta file were produced with a Python (v.3.10.8) script using a biopython (v.1.81) module.

## SL-Smart-seq3xpress library preparation

RNase free reagents were used for all steps and all surfaces were regularly treated with RNaseZAP (Sigma). Each well of a 384-well plate was filled with 3 µl per well silicone oil (Sigma) using an Integra Assist Plus pipetting robot. The plates were sealed with adhesive PCR plate seals and briefly centrifuged to collect the liquid at the bottom of the wells. To each well, 0.3 µl of lysis buffer (0.1% TX-100, 6.67% w/v PEG8000, 0.0417 µM oligo(dT) (5′-Biotin-AGAGACAGATTGCGCAATG[$N_8$][$T_{30}$]VN-3′), 0.67 mM of each dNTP, 0.4 U µl$^{-1}$ RNase inhibitor and spike-in mix (roughly 1,364 transcripts)) were added using an I.DOT liquid dispenser (Cytena). The plates were briefly centrifuged, placed on ice and brought to the cell sorter. Single cells were sorted into each well as described above.

To lyse cells, the reaction plate was thawed, centrifuged and incubated at 72 °C for 10 min. To each well, 0.1 µl of reverse transcription mix (100 mM Tris-HCl pH 8.3, 120 mM NaCl, 10 mM MgCl$_2$, 32 mM DTT, 0.25 U µl$^{-1}$ RNase inhibitor and 8 U µl$^{-1}$ Maxima H-minus Reverse Transcriptase) were immediately added following cell lysis using the I.DOT and the reaction plate then incubated at 42 °C for 90 min before inactivation of the reaction at 85 °C for 5 min. Immediately following the reverse transcription reaction, the reaction plate was centrifuged and 0.6 µl of PCR amplification mix (1.67× SeqAmp PCR buffer, 0.042 U µl$^{-1}$ SeqAmp polymerase, 0.83 µM SL primer: 5′-CTAACGCGTATTATTAGAACAGTTTCTGT*A*C*-3′, and 0.83 µM Reverse primer: 5′-GTCTCGTGGGCTCGGAGATGTGTATAAGAGACAGATCATTGTAGG-3′) were added to each well using the I.DOT liquid dispenser. The PCR reaction was performed with the following conditions: 95 °C for 1 min, 16 cycles of (98 °C for 10 s, 65 °C for 30 s, 68 °C for 4 min), 72 °C for 10 min. Following the reaction, the PCR plate was centrifuged and amplified cDNA was then diluted by adding 9 µl of dH$_2$O to each well of the plate. If not used immediately, the plate was stored at −20 °C until next step.

To each well of a new 384-well plate, 1 µl of each prediluted cDNA was added using the Integra pipetting robot. To each well, 1 µl of tagmentation mix (10 mM Tris-HCl pH 7.5, 5 mM MgCl$_2$, 5% DMF, 0.002 µl of TDE1) was added and the plate incubated at 55 °C for 10 min. To stop the reaction, 0.5 µl of 0.2% SDS was immediately added to each well. The plate was centrifuged and incubated at room temperature for 5 min. The individual libraries generated from each well were dual indexed with Illumina i5 (5′-AATGATACGGCGACCACCGAGATCTACAC[8 bp index]TCGTCGGCAGCGTC-3′) and i7 index primers. For each index primer, 0.5 µl (2.2 µM) was dispensed into each of the reaction wells with 1.5 µl of PCR mix (3.33× Phusion HF Buffer, 0.67 mM of each dNTP, Tween-20 0.083%, and 0.033 U µl$^{-1}$ Phusion HF DNA polymerase), for a final reaction volume of 5 µl. For the sequencing libraries presented in Fig. 3e (right) the volumes were the following: 0.8 µl (2.2 µM) of each index primer and 3.9 µl of PCR mix, for a final reaction volume of 8 µl. The PCR reaction was performed with the following conditions: 72 °C for

3 min, 95 °C for 30 s, 14 cycles of (95 °C for 10 s, 55 °C for 30 s, 72 °C for 1 min), 72 °C for 5 min. The reaction volumes from all wells were pooled into a robotic reservoir (Nalgene, Thermo Scientific) by centrifuging the plate placed in a custom-made three-dimensionally printed plate holder at 200$g$ for 30 s. The pooled library was purified using AMPure XP beads at a ratio of 1:0.7. The libraries were eluted from the beads in 45 µl total volume of dH$_2$O. To further decrease free unligated adaptor concentration, the libraries were run on a 4% non-denaturing PAGE gel and purified according to standard polyacrylamide gel purification protocols. The libraries were sequenced on a NextSeq 1000 sequencing platform to produce paired-end reads of 101 nt (cDNA read) and 19 nt (TAG + UMI read), and 8 nt for the index reads.

## 5′ Chromium 10X library preparation and sequencing

Cultures of N50 and P10 cells[29] were set up and maintained at 0.5–1.0 × 10$^6$ cells per ml before collecting for library preparation. A mixed population sample was prepared by pooling together equal numbers of N50 and P10 cells. The mixed cells were collected by centrifugation at 400$g$ for 10 min, washed twice in ice-cold 1× PBS supplemented with 1% D-glucose and 0.04% BSA, and resuspended in 1 ml of the buffer. The cells were then filtered with a 35 µm cell strainer (Corning) and adjusted to 1,000 cells per µl. Libraries were prepared using the Next GEM Single Cell 5′ GEM Kit v.2 (10xGenomics) and sequenced on the NextSeq 1000 platform to a depth of roughly 50,000 reads per cell. Paired-end reads of 26 nt (read 1) and 122 nt (read 2) as well as 10-nt-index reads were generated.

## Primary processing of SL-Smart-seq3xpress sequencing data

The two reads containing the indexes (8 nt each) and the TAG + UMI (19 nt) were concatenated into a 35 nt read. Artefact reads containing the TAG sequence (or its reverse complement) in the cDNA read were filtered out using Cutadapt[47] (v.4.3). Downsampling of reads for method benchmarking was done using seqtk (v.1.4) 'sample' function (https://github.com/lh3/seqtk). Reads were mapped with STARsolo[48,49] (STAR v.2.7.10a) to a hybrid fasta file combining the *T. brucei* Lister 427 strain genome (Tb427v11, ref. 5) and the spike-in sequences, producing a transcript count matrix and an alignment (BAM) file. The count matrix was then corrected using the index hopping filtering pipeline scSwitchFilter (described in the next section) using the BAM file as input.

## Index hopping filtering

scSwitchFilter (https://github.com/colomemaria/scSwitchFilter) corrects index hopping in multiplexed sequencing libraries using raw BAM files instead of a count matrix[50]. The correction process involves three main steps: (1) BAM to SAM conversion; (2) read extraction and parsing and (3) negative correction count matrix computation. In the first step the pipeline uses samtools[51] (v.1.17) to convert a BAM file to a SAM file. In step 2, a fast bash script is used to extract and parse valid reads from the SAM file, select reads with cell barcode, UMI barcode and gene nametags, and split cell barcode tags for subsequent analysis. The selected reads are then complied into a single .TSV file. Depending on the number of plates (individual libraries) in the sequencing experiment (run), the script may split the cell barcode tag into plate–library-i5-i7 or i5-i7 barcode combinations. In step 3, scSwitchFilter calculates read counts for switched indices, assuming a low probability of the combination of UMI barcode, gene name and an index being present in several plate wells. Reads with more than 80% (default threshold) of total counts among those with switched indices remain unfiltered. The tool generates a residue count matrix that should be subtracted from the initial count matrix to obtain the filtered count matrix.

## SL-Smart-seq3xpress data analysis

Count matrices were processed with JupyterLab (v.4) notebooks using IPython (v.7.31) using the following modules: pandas (v.1.5.3), numpy

(v.1.23.5), scipy (v.1.10.1), scanpy (v.1.7.2), openpyxl (v.3.1.2), matplotlib (v.3.6.3) and seaborn (v.0.12.2). Cells with fewer than 500 genes detected, 1,000 gene UMI transcript counts and 50 spike-in UMI counts were filtered out. For the gene expression analysis, transcript counts for each cell were normalized by spike-in counts. For the quantification of cells expressing each VSG, we defined a cell as expressing a given VSG, if the transcript counts for that VSG represented more than 80% of the transcript counts for all VSGs in that cell. If no VSG reached this threshold, we defined the cell as having 'no dominant VSG'. Final figures were created with Graphpad Prism (v.9).

### Sensitivity and specificity comparison for different sequencing approaches

For Smart-seq2 and SL-Smart-seq3xpress data, reads were subsampled to match the average reads per cell in Chromium 10X (roughly 75,000 reads per cell in Briggs et al.[26] and roughly 100,000 reads per cell for Chromium 10X data from this study). All sequencing data were mapped with STARsolo with identical settings. For the sensitivity comparison, the transcript end coordinate annotations were extended until the beginning of the next transcript, matching the conditions used by Briggs et al. using a perl (v.5.32.1) script with a perl-bioperl (v.1.7.8) module. For specificity comparison, cells with genes detected, total transcript counts or total VSG transcript counts below half of the median of the population of cells, were filtered out. Furthermore, only cells with more than ten VSG UMI counts were considered.

### Type of switching analysis

Single-cell BAM files were extracted from STARsolo output BAM file, using the cell-specific cell barcode:Z attribute (storing the indexes and the TAG sequence) for each mapped read in the BAM file. Only cells with a dominant VSG from 0 h, 96 h and 10 days postinduction time points were considered. Coverage files (Bigwig files) were generated for each single cell using deepTools[52] (v.3.5.4) bamCoverage function with '--normalizeUsing RPKM' and '--minMappingQuality 10' options. Coverage tracks were plotted using pyGenomeTracks[53] (v.3.8). For the determination of switching type (recombination or transcriptional) and for the identification of the transcriptional signal end position in BES1, and the start of transcriptional signal in the BES where the newly active VSG was originally located, the single-cell coverage tracks were visually inspected in Integrative Genomics Viewer (IGV)[54] (v.2.16.0).

### Identification of VSG homologues

Homologues of *VSG-2*, *VSG-8* and *VSG-11* were identified by BLAST (v.2.14.0) to the Lister 427 genome assembly in TriTrypDB[55]. Hits with a bitscore greater than 1,000 were selected as highly similar homologues and putative 'donors' for segmental gene conversion. For *VSG-2*, *VSG-8* and *VSG-11*, there were zero hits, five hits and one hit meeting this criterion, respectively.

### Single-cell de novo VSG transcript assembly after DSB induction in *VSG-8*

Fastq files were demultiplexed into single-cell fastq files with deML[56] (v.1.1.13) with default settings. De novo transcript assemblies were then generated for each single cell with Trinity[57] (v.2.15.1), restricting the output to contigs bigger than 1 kb. To identify which of the assembled transcripts was the active VSG, the de novo assembled contigs were aligned with BLAST[58] (v.2.14.0) to *VSG-8*, and the contigs with high similarity (bitscore greater than 2,000) were extracted using a Python (v.3.10.8) script and reheaded the fasta sequences by cell ID using seqkit (v.2.5.1). Those cells with no contig reaching the threshold were discarded. Multifasta files with all the single-cell de novo assembled VSGs per experiment, together with the putative 'donor' VSGs, were constructed and aligned to VSG-8 with minimap2 (ref. 59) (v.2.10). Finally, the alignments were visualized in IGV and the start and end position of recombination and the putative donor(s) for each cell was determined.

### Bulk RNA-seq library preparation and sequencing

Cell lines expressing different VSGs—*VSG-2*, *VSG-8*, *VSG-11* and those used for the ATAC-seq experiments—were maintained at $0.5–1.0 \times 10^6$ cells per ml before collection. RNA-seq library preparation was performed as previously described[60]. Strand-specific RNA-seq library concentrations were measured in duplicate using Qubit double-stranded DNA HS Assay Kit and Agilent TapeStation system. The libraries were quantified with the KAPA Library Quantification Kit according to the manufacturer's protocol and sequenced on the Illumina NextSeq 1000 platform to generate paired-end reads.

### Bulk RNA-seq data analysis

For the bulk transcriptome analysis of Lister 427 bloodstream form wild-type cells (*VSG-2* expressers) and clones that have switched to the expression of different VSGs (*VSG-8* or *VSG-11*), reads were mapped to the Lister 427 genome assembly v11 with STAR[48] (v.2.7.10a). Coverage files were generated and plotted in the same way as for the scRNA-seq data (section 'Type of switching analysis'). For the analysis of the transcriptional switch time courses, reads were mapped with bwa-mem[61] (v.0.7.17) and PCR duplicates were filtered out with Picard (v.3.2.0) 'MarkDuplicates' function. Counts for each gene were calculated with Subread (v.2.0.1) 'featureCounts' function[62], filtering low confidence mapping reads ('-Q10'). Gene counts were then normalized to kilobases per million mapped reads.

### ATAC-seq library preparation

The ATAC-seq libraries were prepared following the protocol by Müller et al.[10] with several modifications. Briefly, $26.7 \times 10^6$ cells were collected (10 min at 1,800*g*) and washed in 30 ml of ice-cold 1× TDB. The cells were resuspended in 200 µl of permeabilization buffer (100 mM KCl, 10 mM Tris-HCl pH 8.0, 1 mM DTT, 25 mM EDTA) supplied with protease inhibitors. After adding 2 µl of 4 mM digitonin, the cells were incubated for 5 min at room temperature. Next, the cells were pelleted at 1,200*g* for 10 min at 4 °C, resuspended in 400 µl of isotonic buffer (100 mM KCl, 10 mM Tris-HCl pH 8.0, 10 mM $CaCl_2$, 5% glycerol) with protease inhibitors and pelleted again. Tagmentation was performed by adding 50 µl of tagmentation mix (25 µl of 2× reaction buffer, 24 µl of $dH_2O$, 1 µl TDE1) to the cell pellet and incubating at 37 °C for 30 min. The DNA was then purified using Qiagen MinElute PCR Purification Kit, eluted in 10 µl of elution buffer (10 mM Tris-HCl, pH 8.0), and amplified for 13 cycles using Phusion High-Fidelity DNA Polymerase with 2.5 µl of index primers (each, 25 mM) in a 50 µl of reaction mixture. The resulting libraries were purified using AMPure XP beads at a 1.8× ratio and eluted in 20 µl of nuclease-free water. The libraries were sequenced on a NextSeq 1000 platform to generate paired-end reads of 60 nt each to a depth of 400 million reads.

### ATAC-seq data analysis

Reads were mapped to the Lister 427 genome assembly v.11 with bwa-mem[61] (v.0.7.17). Counts per million normalized coverage per 25-nt bin was calculated with 'bamCoverage', whereas filtering reads with low mapping quality ('-Q10'). The average coverage for each BESs was calculated with 'multibigSummary' function from deepTools[52] (v.3.5.1). For each BES and sample, the $\log_2$ ratio relative to the initial silent state was calculated.

### BLISS

BLISS was performed as previously described[63] (with the modifications described below). Furthermore, starting cell concentration was adjusted according to previous BLISS experiments in trypanosomes[39]. Cas9 was induced by incubating $2 \times 10^8$ cells with doxycycline for 4 h before cell collection. Cells were pelleted for 10 min at 800*g* and resuspended in 17.5 ml of warm 1× TDB, followed by fixation in 2% methanol-free formaldehyde for 10 min at room temperature

with rotation. Formaldehyde was quenched by addition of glycine to a final concentration of 125 mM and incubation with rotation for 5 min at room temperature and 5 min on ice. Crosslinked cells were pelleted and washed in 20 ml of ice-cold 1× TDB, transferred to a 1.5 ml of protein LoBind tube (Eppendorf) and washed again with ice-cold 1× TDB. Crosslinked cells were counted using a Neubauer chamber and kept at 4 °C for up to two weeks before starting the BLISS template preparation. Next, $5 \times 10^7$ crosslinked cells were lysed in 200 µl of lysis buffer 1 (10 mM Tris pH 8.0, 10 mM NaCl, 1 mM EDTA, 0.2% Triton X-100) for 1 h on ice and pelleted and incubated in 200 µl of prewarmed lysis buffer 2 (10 mM Tris pH 8.0, 150 mM NaCl, 0.3% SDS) for 1 h at 37 °C with shaking at 400 rpm. The cells were washed twice with 200 µl of prewarmed CSTX buffer (1× rCutSmart buffer with 0.1% Triton X-100). DSB blunting of the sample was performed using the Quick Blunting Kit (NEB) for 1 h at 25 °C with shaking at 400 rpm and followed by two washes with 200 µl of CSTX buffer. Four microlitres of sample-specific 10 µM annealed BLISS adaptors were ligated to the blunted DSBs for 20 h at 16 °C using T4 DNA ligase (Thermo Fisher). Ligated samples were washed twice with 200 µl of CSTX buffer before resuspension in 100 µl of Tail buffer (10 mM Tris pH 7.5, 100 mM NaCl, 50 mM EDTA, 0.5% SDS) with 10 µl of proteinase K (NEB, 800 U ml$^{-1}$). Samples were incubated overnight at 55 °C with shaking at 800 rpm, followed by the addition of another 10 µl of Proteinase K (NEB, 800 U ml$^{-1}$) and incubation was continued for another hour. Proteinase K was deactivated by incubating the samples at 95 °C for 10 min. The template DNA was extracted using phenol–chloroform–isoamyl alcohol mixture, followed by ethanol precipitation and eluted in 130 µl of TE buffer (10 mM Tris pH 8.0, 1 mM EDTA). The extracted template DNA was sonicated in microtubes for 80 s using a Covaris S220 and the following settings: duty factor 10%, PIP 140 W, 200 cycles per burst. Sheared DNA was then purified with 0.8× AMPure XP beads and eluted in 15 µl of nuclease-free water and analysed on the TapeStation. To prepare BLISS libraries, 50–100 ng of the purified DNA were used for in vitro transcription of the template DNA using the MEGAscript T7 Transcription Kit (Thermo Fisher) and a sample incubation of 15 h at 37 °C. Next, the DNA template was degraded with Turbo DNase I (Thermo Fisher) and the amplified RNA was purified with 1× AMPure XP beads and eluted in 6 µl of nuclease-free water. The amplified RNA was analysed on the TapeStation. Afterwards, 1 µl of the 10 µM RA3 adaptor was ligated to the purified amplified RNA using T4 RNA Ligase 2 truncated (NEB), followed by reverse transcription with 2 µl of the 10 µM RTP and Superscript IV Reverse Transcriptase (Thermo Fisher). Libraries were indexed and amplified by PCR with NEBNext Ultra II Q5 (NEB) using a library-specific RPIX indexed primer and the RP1 common primer. The PCR was performed by splitting each sample into eight different PCR tubes. Double-sided clean-up was performed on the libraries with 0.45–0.75× AMPure XP beads. Final libraries were analysed on TapeStation and the library pool was quantified using the KAPA Library Quantification Kit (Roche). The libraries were sequenced with paired-end (80 cycles for read 1 and 52 cycles for read 2) on a Next-Seq1000 (Illumina) sequencing platform to a depth of 40 million reads per library.

### BLISS data analysis

Forward reads were trimmed to remove library barcodes and UMIs using Cutadapt[47] (v.3.5), allowing up to one mismatch. The trimmed sequences were then added to their respective read names using an available Python script[39]. Furthermore, to avoid cross-mapping of guide RNA (gRNA)-derived sequences on the target position, reads containing the gRNA scaffold were discarded with Cutadapt. Reads were then aligned to the Lister 427 genome assembly v.11 with bwa-mem[61] (v.0.7.17) with default parameters. Aligned reads were then deduplicated using the UMI on the header of the forward read with the function 'dedup' of umi_tools[64] (v.1.1.2). Filtered alignment files from replicate experiments were merged with samtools[51] (v.1.20) 'merge' function and the normalized coverage for the start of the forward reads, marking the

DSB positions, in 10-nt bins, was calculated with bamCoverage function from deepTools[52] (v.3.5.1) with the following parameters: '--binSize 10', '--Offset 11' '--samFlagInclude 66' and '--normalizeUsing CPM'. Finally, the ratio coverage to the one in Lister 427 bloodstream form wild-type cells was calculated with 'bigwigCompare' function from deepTools, and plotted for specific regions with pyGenomeTracks[53] (v.3.8).

### Reporting summary

Further information on research design is available in the Nature Portfolio Reporting Summary linked to this article.

### Data availability

The scRNA-seq, RNA-seq, ATAC-seq and BLISS data generated for this project have been deposited in the European Nucleotide Archive and are accessible through ENA study accession number PRJEB72370. The scRNA-seq data published by Müller et al.[10] used in this project are deposited in the Gene Expression Omnibus (GEO) and are accessible through GEO Series accession number GSE100896. The Tb427v11 genome assembly is available at Zenodo (https://doi.org/10.5281/zenodo.10692100)[65]. Source data are provided with this paper.

### Code availability

Workflows and custom-made Unix Shell and Python scripts to analyse the data are available at Zenodo (https://doi.org/10.5281/zenodo.10692100)[65].

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

**Acknowledgements** We thank S. Esser and E. Osses for help with the preparation of scRNA-seq and RNA-seq libraries and all members of the Siegel laboratory and the Divisions of Physiological Chemistry and Experimental Parasitology for valuable discussion. We thank the Laboratory for Functional Genome Analysis (LAFUGA) at the Gene Center Munich, LMU, for next-generation sequencing, the Core Facility Flow Cytometry at the Biomedical Center, LMU, for providing equipment and expertise, and the Biomedical Center (BMC) Core Facility Bioinformatics for providing access to the computing server. We thank A.-E. Saliba (Institute for RNA-based Research, Würzburg Germany) and M. Hagemann-Jensen from the Sandberg Laboratory (Karolinska Institutet, Stockholm, Sweden) for advice on the adaption of the Smart-seq3xpress protocol. We thank L. Glover (Institut Pasteur, Paris, France) for providing the γH2A antibody, E. Beltrán (BMC, LMU) for 10X Genomics sequencing and A. Taddei (Institut Curie, Paris, France) for sharing valuable insights regarding DSB repair mechanisms. This work was funded by the German Research Foundation (SI 1610/2-2 and 213249687—SFB 1064 to T.N.S.), and a European Research Council (ERC) Starting grant (no. 3D_Tryps 715466) and an ERC Consolidator grant (no. SwitchDecoding 101044320) awarded to T.N.S. Z.K. and A.D. were supported by MSCA ITN Cell2Cell (grant no. 86067) fellowships. K.R.M. was supported by the European Union's Framework Programme for Research and Innovation Horizon 2020 (grant no. 2014-2020) under the Marie Skłodowska-Curie grant agreement no. 754388 (LMUResearchFellows) and from LMUexcellent, funded by the Federal Ministry of Education and Research (BMBF) and the Free State of Bavaria under the Excellence Strategy of the German Federal Government and the Länder, and the Wellcome Trust (grant no. 221717/Z/20/Z).

**Author contributions** The experiments were designed by Z.K., K.R.M., R.O.C., I.S., M.C.-T. and T.N.S. and carried out by Z.K., K.R.M., R.O.C. and I.S., unless otherwise indicated. Cell lines were generated by K.R.M. and I.S. The SL-Smart-seq3xpress approach was developed by Z.K., K.R.M., R.O.C. and T.N.S. SL-Smart-seq3xpress libraries were generated by Z.K. and R.O.C. BLISS libraries were produced by A.B.-S. Computational analyses were performed by R.O.C. and A.D. Z.K., K.R.M., R.O.C., I.S., A.B.-S., A.D., J.E.S., M.R.M., M.C.-T. and T.N.S. contributed to the data interpretation and development of a model. The work was supervised by M.C.-T. and T.N.S. The manuscript was written by K.R.M. and T.N.S. with help from Z.K. and R.O.C. and edited by all other co-authors. The figures were generated by Z.K., K.R.M. and R.O.C.

**Competing interests** The authors declare no competing interests.

**Additional information**
**Correspondence and requests for materials** should be addressed to Maria Colomé-Tatché or T. Nicolai Siegel.

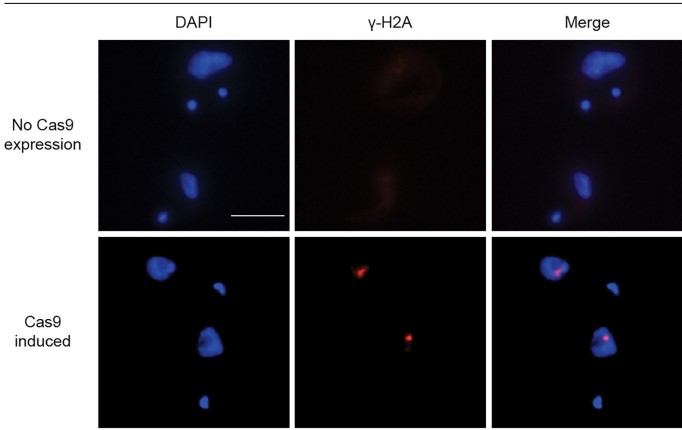

**Extended Data Fig. 1 | γ-H2A IFA before and after Cas9 induction.** IFA of
γ-H2A expression in cells transfected with sgRNA VSG-2.1140 either before
(No Cas9 expression) or 4 h after Cas9 induction (Cas9 induced). γ-H2A was
detected with an anti-γ-H2A antibody. Scale bar, 10 μm; n = 2.

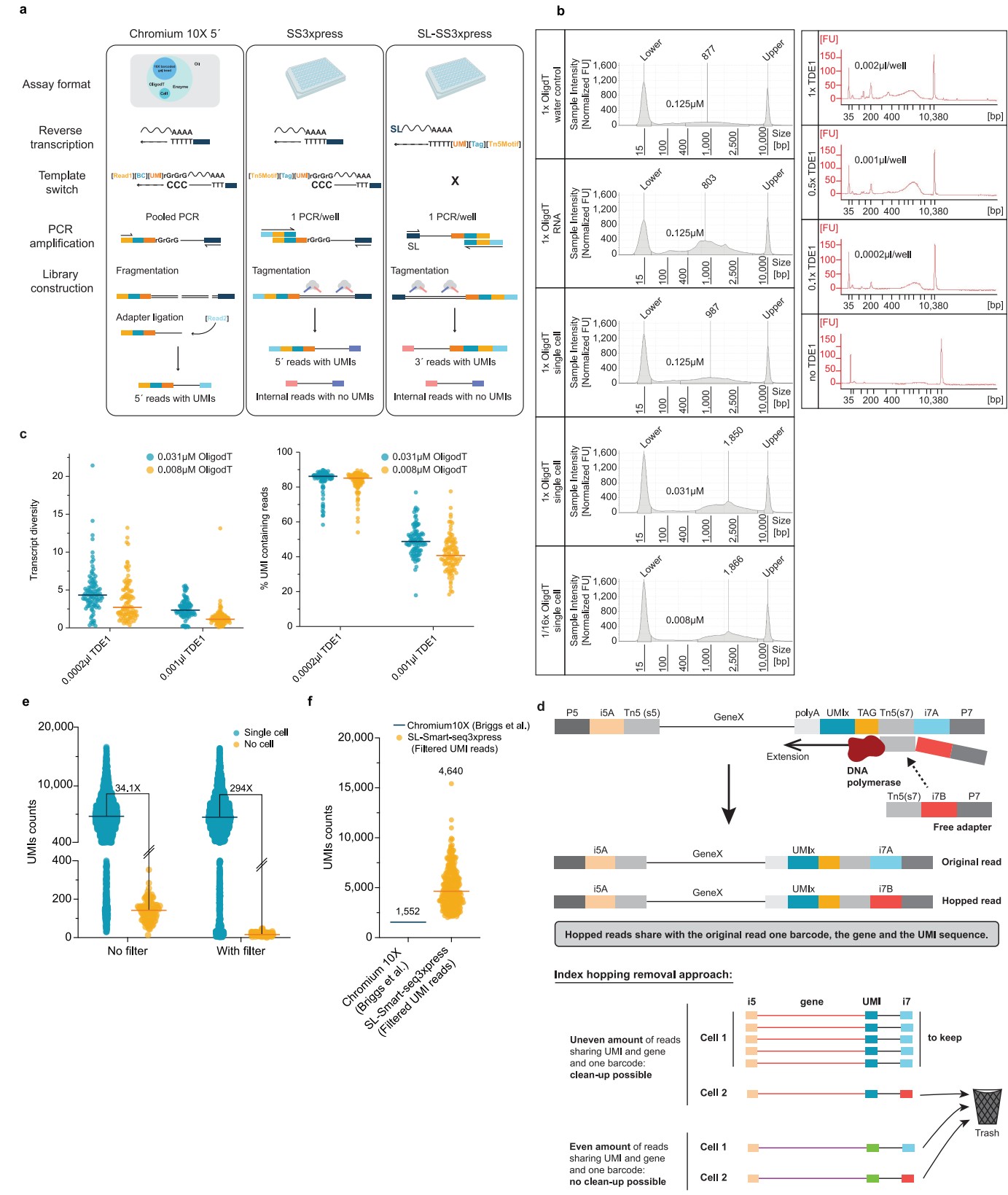

**Extended Data Fig. 2** | See next page for caption.

**Extended Data Fig. 2 | Optimization of the SL-Smart-seq3xpress scRNA-seq pipeline. a**. The library preparation workflows for: the Chromium 10 × 5′ pipeline (left panel); the standard Smart-seq3xpress (SS3xpress) pipeline (middle); and the tailored SL-Smart-seq3xpress (SL-SS3xpress) pipeline (right). BC, barcode; SL, spliced leader. **b**, Left panel: Representative TapeStation profiles of single-cell libraries (pooled and bead-purified after cDNA dilution) prepared using different oligo(dT) concentrations - 1X, 1/4X, 1/16X - relative to the concentration in the published Smart-seq3xpress protocol[20]. Each condition was tested with at least two independent replicates, each containing between 6 and 48 cells. Right panel: TDE1 Tn5 concentrations -1X, 0.5X, 0.1X, relative to the concentration in the published Smart-seq3xpress protocol[20] - tested to optimize tagmentation. Here, each condition contains 48 cells. **c**, Optimization of SL-Smart-seq3xpress oligo(dT) and TDE1 enzyme concentrations. Left panel: Transcript diversity of SL-Smart-seq3xpress libraries prepared with varying oligo(dT) and TDE1 concentrations. Transcript diversity was measured as percentage of UMI counts/per reads sequenced/single cell. Right panel: Percentage of UMI-containing reads in SL-Smart-seq3xpress libraries prepared with varying oligo(dT) and TDE1 concentrations. Each dot represents a single cell. 96 cells were sequenced for each condition tested. **d**, Schematic of the mechanism leading to index hopping and our bioinformatic strategy to remove index hopped reads. **e**, Number of UMIs detected by SL-Smart-seq3xpress in single cells (blue) or no cell wells (orange) either without (No filter) or with (With Filter) bioinformatic removal of index hopped reads. The ratio of median UMI counts (single cell/no cell) is shown as a number between the two conditions. The median UMI count is shown as a black line. Number of wells analyzed: single cells - 2214, no cells - 120. **f**, Comparison of the median number of UMIs detected by Chromium 10X (data from Briggs et al.[26]) or SL- Smart-seq3xpress. Elements of this figure have been created in BioRender.

**a**

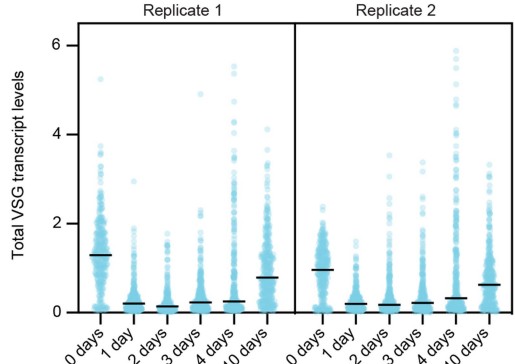

**b**

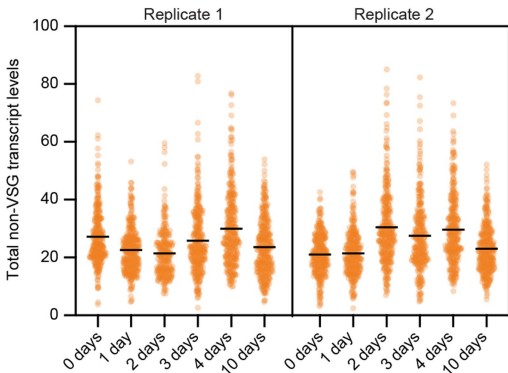

**c**

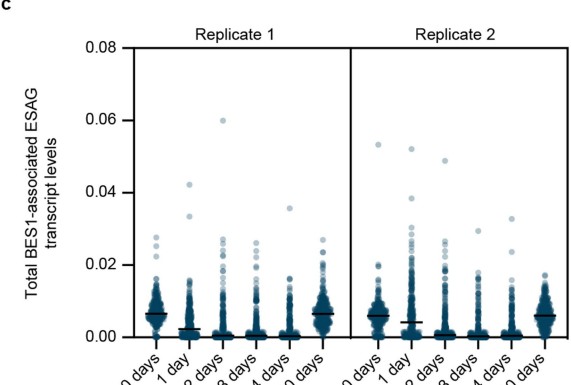

**d**

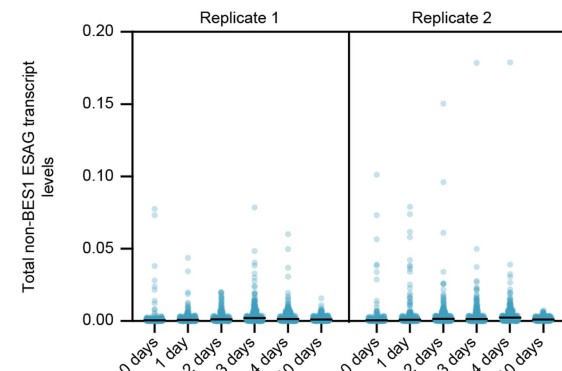

**Extended Data Fig. 3 | Relative expression levels of VSGs and ESAGs during DSB induction at nucleotide position 1140 of the *VSG-2* CDS. a-d**, Single cell transcript levels normalized to spike-in counts. Data is shown separately for each of the two biological replicates. **a**, Total VSG genes, **b**, Total non-VSG genes, **c**, Total BES1 ESAGs and **d**, Total non-BES1 ESAGs.

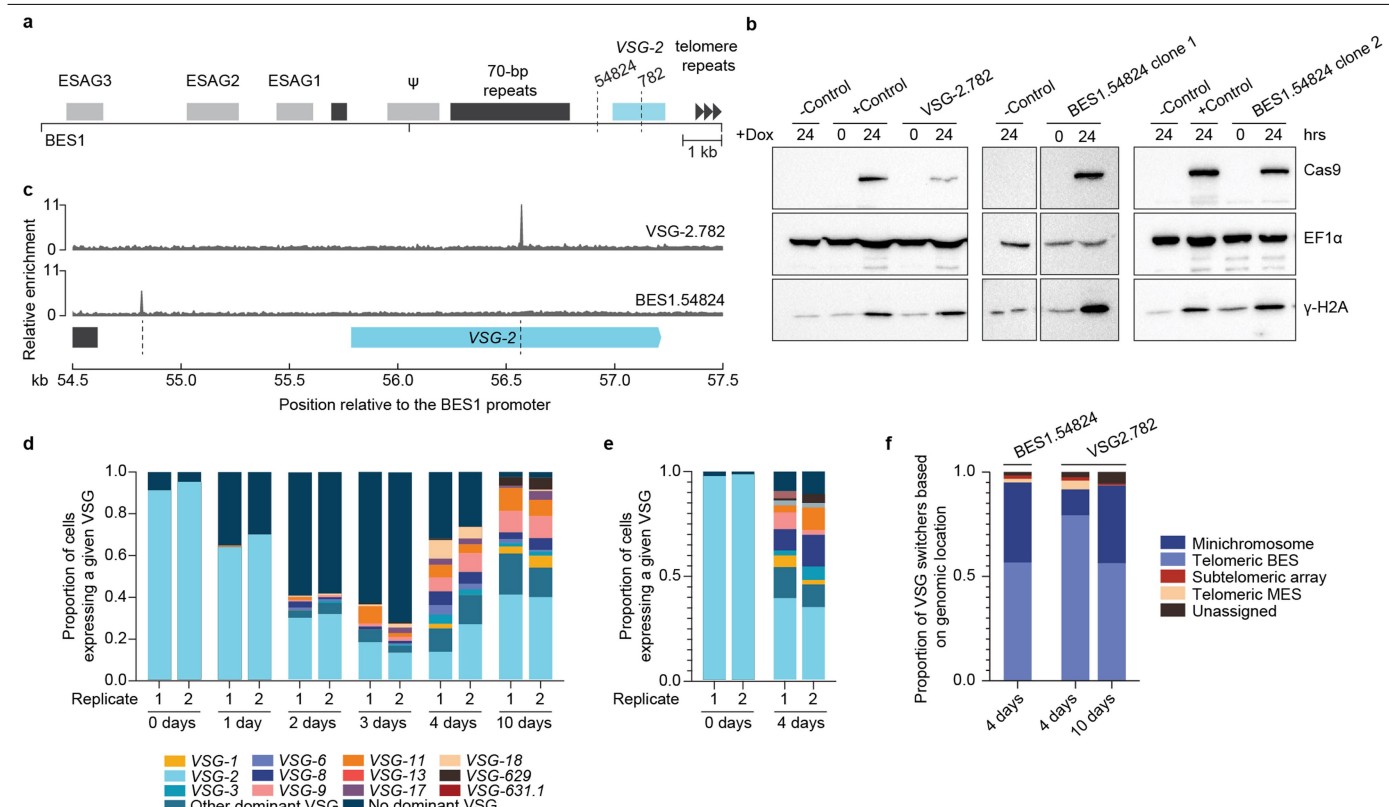

**Extended Data Fig. 4 | DSBs in or just upstream of *VSG-2* lead to a very similar switching profile outcome. a**, Schematic map of the terminal region of BES1 with the cut sites (dashed lines). The coordinates are relative to the promoter or to the start of the *VSG-2* CDS. **b**, Western blot showing Cas9 and γ-H2A expression levels before and 24 h post-Cas9 induction with doxycycline. Wild-type Lister 427 cells 24 h after doxycycline addition were used as a negative control. EF1α was used as a loading control. Clones used for scRNA-seq experiments are highlighted in bold (n = 1). **c**, BLISS coverage tracks on BES1 after 4 h of Cas9 induction in the cell lines with sgRNA VSG-2.782 and BES1.54824, both normalized to the BLISS coverage of wild-type cells. The light blue box represents the *VSG-2* CDS and the black box represents the 70-bp

repeats. The on-target DSB position for each cell line is indicated by a dashed line. Shown is the average of two biological replicates. **d**, scRNA-seq results of the time course experiment following DSB induction at nucleotide position 782 in the *VSG-2* CDS, showing the proportion of cells expressing a given VSG at each time point for each biological replicate. **e**, scRNA-seq results before and 4 days after DSB induction at position 54,824 of BES1, showing the proportion of cells expressing a given VSG at each time point for each biological replicate. **f**, For the experiments in **(d)** and **(e)**, proportion of switcher cells grouped by the genomic location of the newly activated VSG. Cells expressing a VSG for which the genomic location is unknown were categorized as 'unassigned'. For gel source data, see Supplementary Fig. 1a.

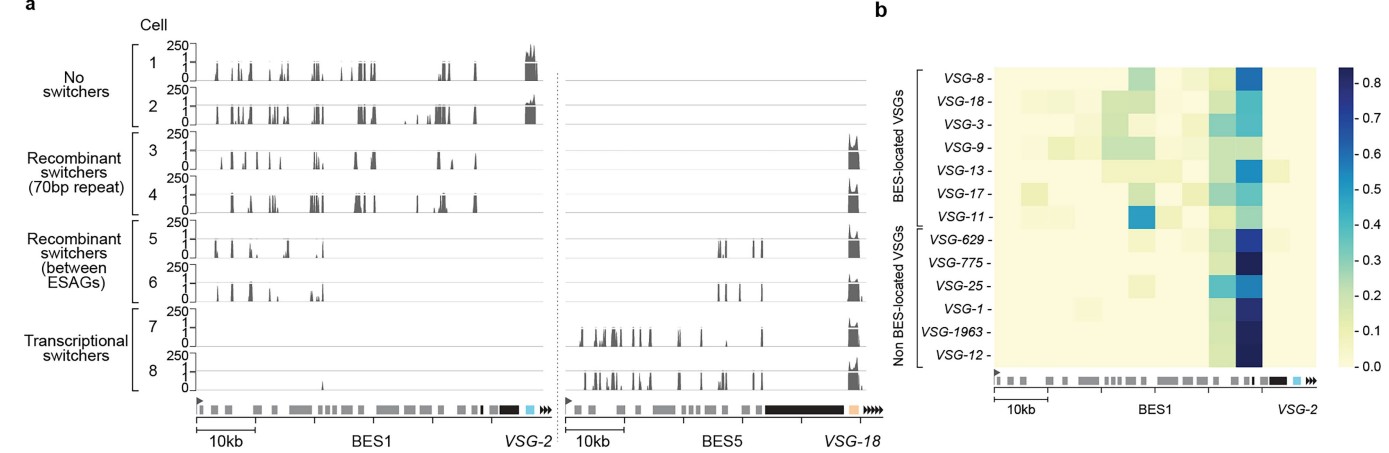

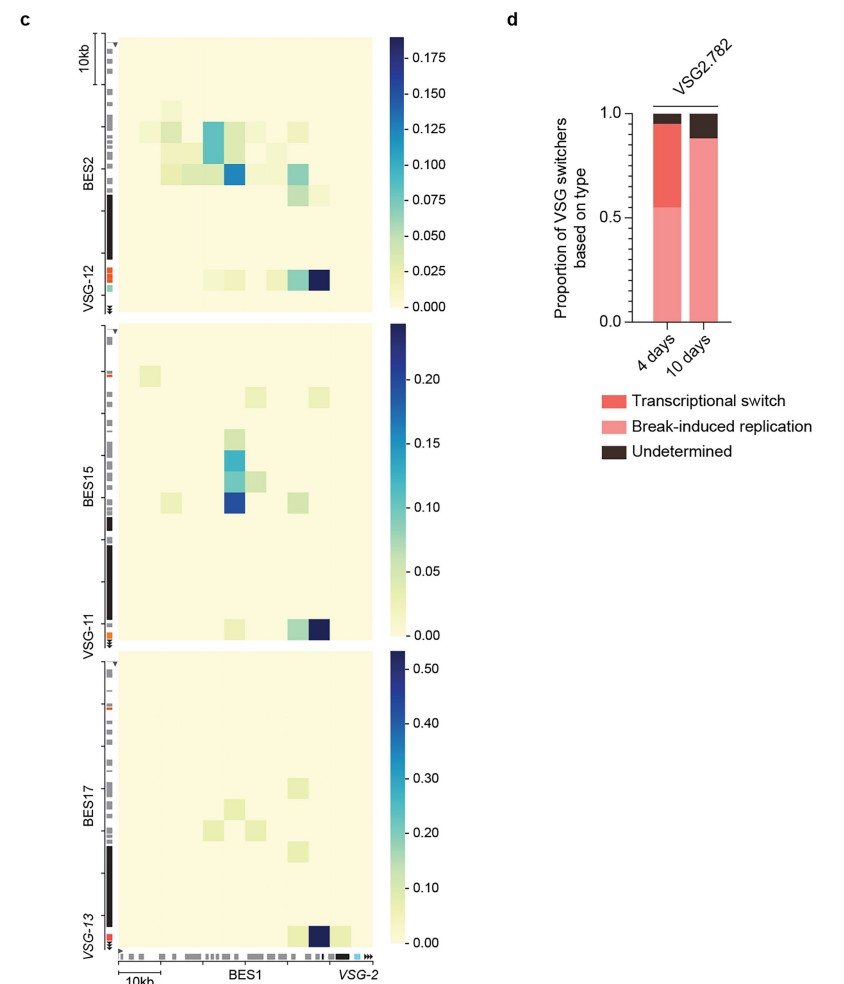

**Extended Data Fig. 5** | See next page for caption.

**Extended Data Fig. 5 | Heat map of putative recombination sites in BES1.**
**a**, Single-cell sequencing read coverage on BES1 and BES5 showing different VSG switching scenarios for cells that switched to *VSG-18* expression following a DSB in the *VSG-2* CDS. Read coverage is shown for 8 representative cells, 2 cells per VSG switching scenario: no switching (No switchers, cells expressing *VSG-2*), switching by recombination around the 70-bp repeats (Recombinant switchers (70-bp repeat)), switching by recombination between ESAGs (Recombinant switchers (between ESAGs)), and transcriptional switching (Transcriptional switchers). **b** and **e**, Heatmaps summarizing the transcriptional signal end positions on BES1 for recombinant switcher cells after a DSB at nucleotide position 1140 **(b)** and 782 **(e)** of the *VSG-2* CDS. The color of each of the squares in the heatmaps represents the fraction of cells, for switchers to a given VSG (rows), for which transcriptional signal ends at the given 5 kb bin of BES1 (columns). VSGs expressed in at least 10 cells were considered. **c**, Heatmaps summarizing transcriptional signal end position on BES1 (columns) for recombinant switcher cells versus the transcriptional signal start on the BES containing the incoming VSG (rows), after a DSB in the active *VSG-2* CDS (at nucleotide position 1140). Each square on the heatmap represents a 5 kb bin of BES1 (columns) and the BES from the incoming VSG (rows). The color of each square represents the fraction of cells for which transcriptional signal ends at that given bin of BES1 and starts at that given bin of the BES containing the incoming VSG. The heatmaps summarize the data from 122 cells switching to *VSG-9*, 41 cells switching to *VSG-11* and 15 cells switching to *VSG-13*, at 4 days and 10 days after DSB induction. **d**, Proportion of cells that had switched VSG expression by recombination or transcriptional switch at 4 days and 10 days after DSB induction at nucleotide position 782 in the *VSG-2* CDS. Cells for which the switching mechanism could not be clearly determined, are labeled as 'undetermined'.

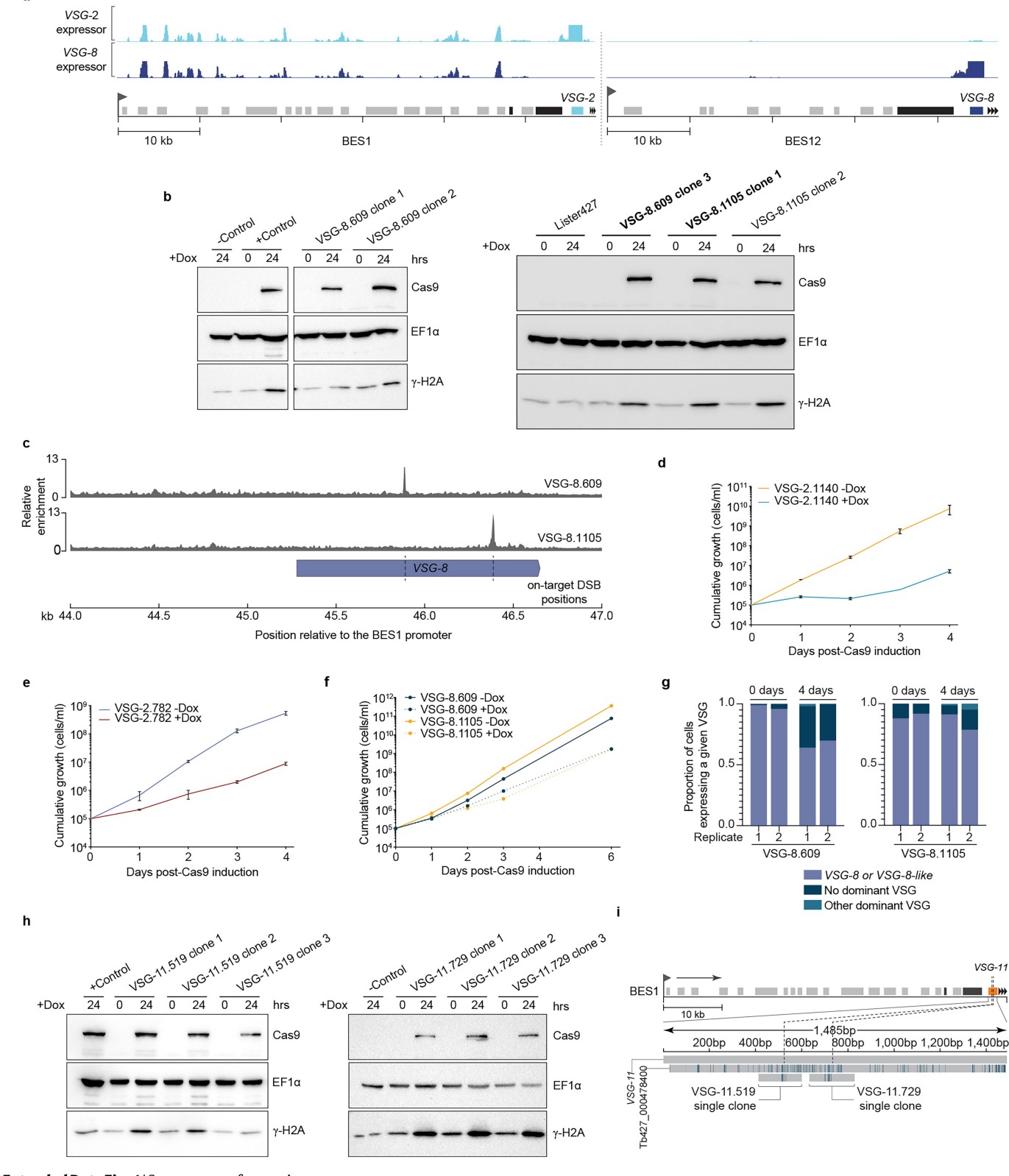

**Extended Data Fig. 6** | See next page for caption.

**Extended Data Fig. 6 | DSBs in VSGs with homologous sequences in the genome lead to segmental gene conversion. a**, Read coverage from bulk RNA-seq data on BES1 (left panel) and BES12 (right panel) for a control *VSG-2* expressing cell line and the *VSG-8* expressing cell line cloned from a DSB induction experiment at nucleotide position 1140 of the *VSG-2* CDS; showing that in this cell line *VSG-8* was recombined and is being expressed from BES1 (shown is a representative profile of one out of three biological replicates). **b**, Western blot showing Cas9 and γ-H2A expression levels before and 24 h after Cas9 induction with doxycycline (Dox) for two cell lines – sgRNA VSG-8.609 (clones 1 and 2, left panel, clone 3, right panel) and sgRNA VSG-8.1105 (clones 1 and 2, right panel) - expressing different sgRNAs to generate DSBs at positions 609 and 1105 of the *VSG-8* CDS, respectively. The clones used for scRNA-seq experiments are shown in bold. In the left panel, the negative control is the wild-type cell line and the positive control is the sgRNA VSG2.1140 cell line (n = 1). **c**, BLISS coverage tracks on BES1 after 4 h Cas9 induction in the sgRNA VSG-8.609 and sgRNA VSG-8.1105 cell lines, both normalized to BLISS coverage of wild-type cells. The light blue box represents the *VSG-2* CDS and the black box represents the 70-bp repeats. The on-target DSB position for each cell line is indicated with a dashed line. Shown is the average of two biological replicates. **d-f**, Growth curves following Cas9-based DSB induction for cut sites at **(d)**

nucleotide position 1140 of *VSG-2* CDS, shown is the mean ± SD of three biological replicates, **(e)** nucleotide position 782 of *VSG-2* CDS, shown is the mean ± SD of three biological replicates and **(f)** and nucleotide positions 609 and 1105 of *VSG-8* CDS, the values are derived from a single experiment. **g**, Proportion of cells expressing a given VSG before and 4 days after DSB induction at nucleotide positions 609 and 1105 of the *VSG-8* CDS. Data are shown for two replicates per time point. **h**, Western blot showing Cas9 and γ-H2A expression levels before and 24 h after Cas9 induction with doxycycline (Dox) for two cell lines – sgRNA VSG-11.519 (clones 1,2 and 3, left panel) and sgRNA VSG-11.729 (clones 1,2 and 3, right panel) - expressing different sgRNAs to generate DSBs at positions 519 and 729 of the *VSG-11* CDS, respectively. Clones used for scRNA-seq experiments are shown in bold. The positive control is the sgRNA VSG2.1140 cell and the negative control is the wild-type cell line (n = 1). **i**, Sanger sequencing results for clones derived from a *VSG-11* expressing cell line after DSB induction at nucleotide positions 519 and 729 of the *VSG-11* CDS. A single clone is shown for each cut. Sanger sequences are aligned against the reference *VSG-11*, together with a potential 'donor' sequence (Tb427_000478400:pseudogene, Tb427v11 genome). For gel source data, see Supplementary Fig. 1a.

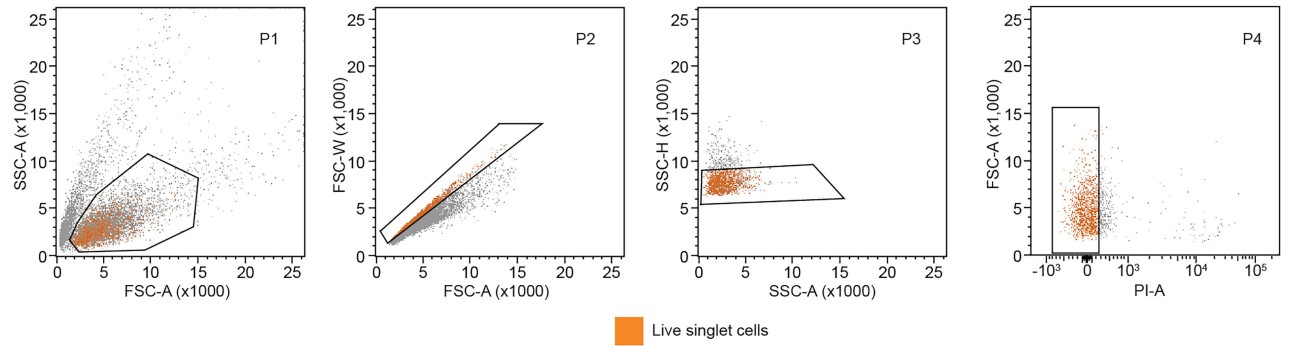

Live singlet cells

**Extended Data Fig. 7 | FACS gating strategy for SL-Smart-seq3xpress library preparation.** The cells were gated to exclude debris (P1), doublets (P2 and P3), and dead cells (P4). Propidium iodide was used to gate for live cells. The panel shows that P4 (colored orange) was used as a final cell population for sorting into the 384-well plates.

# Reporting Summary

## Statistics

For all statistical analyses, confirm that the following items are present in the figure legend, table legend, main text, or Methods section.

| n/a | Confirmed | |
|---|---|---|
| ☐ | ☒ | The exact sample size (*n*) for each experimental group/condition, given as a discrete number and unit of measurement |
| ☐ | ☒ | A statement on whether measurements were taken from distinct samples or whether the same sample was measured repeatedly |
| ☒ | ☐ | The statistical test(s) used AND whether they are one- or two-sided *Only common tests should be described solely by name; describe more complex techniques in the Methods section.* |
| ☒ | ☐ | A description of all covariates tested |
| ☒ | ☐ | A description of any assumptions or corrections, such as tests of normality and adjustment for multiple comparisons |
| ☐ | ☒ | A full description of the statistical parameters including central tendency (e.g. means) or other basic estimates (e.g. regression coefficient) AND variation (e.g. standard deviation) or associated estimates of uncertainty (e.g. confidence intervals) |
| ☒ | ☐ | For null hypothesis testing, the test statistic (e.g. *F*, *t*, *r*) with confidence intervals, effect sizes, degrees of freedom and *P* value noted *Give P values as exact values whenever suitable.* |
| ☒ | ☐ | For Bayesian analysis, information on the choice of priors and Markov chain Monte Carlo settings |
| ☒ | ☐ | For hierarchical and complex designs, identification of the appropriate level for tests and full reporting of outcomes |
| ☒ | ☐ | Estimates of effect sizes (e.g. Cohen's *d*, Pearson's *r*), indicating how they were calculated |

*Our web collection on statistics for biologists contains articles on many of the points above.*

## Software and code

Policy information about availability of computer code

| Data collection | Software used for data collection (Immunofluorescence analysis): LASX  / v3.7.6 software / https://www.leica-microsystems.com/products/microscope-software/p/leica-las-x-ls/ |
|---|---|
| Data analysis | SOFTWARE / VERSION<br>FlowJoTM Software / v10.10.0<br>Graphpad / v9<br>Fiji / v2.0<br>python / v3.10.8<br>IPython / v7.31<br>deepTools / v3.5.4<br>STAR / v2.7.10a<br>BLAST / v2.14.0<br>minimap2 / v2.10<br>seqkit / v2.5.1<br>perl / v.5.32.1<br>perl-bioperl / v1.7.8<br>trinity / v2.15.1<br>seqtk / v1.4<br>cutadapt / v4.3<br>cutadapt / v3.5<br>biopython / v1.81 |

```
jupyterlab / v4
matplotlib / v3.6.3
pandas / v1.5.3
numpy / v1.23.5
seaborn / v0.12.2
pygenometracks / v3.8
scanpy / v1.7.2
openpyxl / v3.1.2
scipy / v1.10.1
scSwitchFilter / v1.0.0
bwa / v0.7.17
picard / v3.2.0
umi_tools /v1.1.2
subread /v2.0.1
samtools / v1.17
samtools / v1.20
Protospacer Workbench / v0.1.0 beta
ChemiDoc MP Imaging System / v3.0.1.14
IGV / v2.16.0
deML / v1.1.13
FCS Express software / v7

Hardware:
High performance computing system (HPC) from the Biomedical Center Munich ( https://www.compbio.bmc.med.uni-muenchen.de/hpc/
index.html )

Operating system:
CentOS 7; with GNU bash, version 4.2.46(1)-release (x86_64-redhat-linux-gnu). Processes were run using SLURM (job scheduling system)
version v16.05.2

Workflows and scripts code:
All custom scripts and computational workflows are publicly available at Zenodo (https://doi.org/10.5281/zenodo.10692101). Computational
environment files (conda yaml files) to reproduce the software set up used during the analysis are provided. Documentation for reproducing
data analysis is provided.
```

For manuscripts utilizing custom algorithms or software that are central to the research but not yet described in published literature, software must be made available to editors and reviewers. We strongly encourage code deposition in a community repository (e.g. GitHub). See the Nature Portfolio guidelines for submitting code & software for further information.

# Data

Policy information about availability of data

All manuscripts must include a data availability statement. This statement should provide the following information, where applicable:
- Accession codes, unique identifiers, or web links for publicly available datasets
- A description of any restrictions on data availability
- For clinical datasets or third party data, please ensure that the statement adheres to our policy

Data availability
The scRNA-seq, RNA-seq, ATAC-seq and BLISS data generated for this project have been deposited in the European Nucleotide Archive and are accessible through ENA study accession number PRJEB72370. The scRNA-seq data published by Müller et al. (doi.org/10.1038/s41586-018-0619-8) used in this project is deposited in the Gene Expression Omnibus and are accessible through GEO Series accession number GSE100896. The Tb427v11 genome assembly is available in Zenodo (https:// doi.org/10.5281/zenodo.10692100).

# Research involving human participants, their data, or biological material

Policy information about studies with human participants or human data. See also policy information about sex, gender (identity/presentation), and sexual orientation and race, ethnicity and racism.

| | |
|---|---|
| Reporting on sex and gender | N/A |
| Reporting on race, ethnicity, or other socially relevant groupings | N/A |
| Population characteristics | N/A |
| Recruitment | N/A |
| Ethics oversight | N/A |

Note that full information on the approval of the study protocol must also be provided in the manuscript.

# Field-specific reporting

Please select the one below that is the best fit for your research. If you are not sure, read the appropriate sections before making your selection.

☒ Life sciences ☐ Behavioural & social sciences ☐ Ecological, evolutionary & environmental sciences

For a reference copy of the document with all sections, see nature.com/documents/nr-reporting-summary-flat.pdf

# Life sciences study design

All studies must disclose on these points even when the disclosure is negative.

| | |
|---|---|
| Sample size | Sample size was not statistically predetermined for the individual experiments. The sample size selected offered a good compromise between scale and cost. Due to the low heterogeneity the number of cells analyzed were sufficient for the analysis. |
| Data exclusions | SL-Smart-seq3xpress data analysis: Cells with less than 500 genes detected, 1000 gene UMI transcript counts were filtered-out. |
| Replication | All attempts of replication were successful. All scRNA-seq experiments were done in biological duplicates. Cell density measurements were done in triplicates except for growth curves shown in Extended Data Figure 6f, for which only one measurement was taken. No experiments other than those mentioned in the reporting summary were performed. |
| Randomization | No randomization applied. Since cell sorting was done in different days no randomization was possible. |
| Blinding | No blinding applied, as we do not perform case / control analyses. |

# Reporting for specific materials, systems and methods

We require information from authors about some types of materials, experimental systems and methods used in many studies. Here, indicate whether each material, system or method listed is relevant to your study. If you are not sure if a list item applies to your research, read the appropriate section before selecting a response.

## Materials & experimental systems

| n/a | Involved in the study |
|---|---|
| ☐ | ☒ Antibodies |
| ☐ | ☒ Eukaryotic cell lines |
| ☒ | ☐ Palaeontology and archaeology |
| ☒ | ☐ Animals and other organisms |
| ☒ | ☐ Clinical data |
| ☒ | ☐ Dual use research of concern |
| ☒ | ☐ Plants |

## Methods

| n/a | Involved in the study |
|---|---|
| ☒ | ☐ ChIP-seq |
| ☐ | ☒ Flow cytometry |
| ☒ | ☐ MRI-based neuroimaging |

# Antibodies

| | |
|---|---|
| Antibodies used | ANTIBODY / SOURCE / IDENTIFIER<br>Alexa Fluor TM 488-conjugated anti-VSG-2 / The antibody was generated by Pinger et al, 2017 (https://doi.org/10.1038/s41467-017-00959-w), acquired from the Memorial Sloan Kettering, Antibody & Biosource Core Facility (https://www.mskcc.org) and conjugated using the Alexa FluorTM 488 Antibody Labeling Kit (A10235)<br>Mouse anti-CRISPR/ Cas9 7A9-3A3 /Active Motif / Cat # 61978<br>Rabbit anti-gamma-H2A / Dr. Lucy Glover, Institut Pasteur / Glover and Horn, 2012 (doi: 10.1016/j.molbiopara.2012.01.008)<br>Mouse anti-EF1α CBP-KK1 / Merck-Millipore / Cat# 05-235 RRID:AB_309663<br>Goat anti-mouse HRP (used for anti-Cas9 and anti-EF1alpha) / GE Healthcare/ code NA931V<br>Goat anti-rabbit HRP (used for anti-gammaH2A)/ GE Healthcare/ code NA934V |
| Validation | There was no new antibody generated for this study. All antibody validations have been performed previously, see above. |

# Eukaryotic cell lines

Policy information about cell lines and Sex and Gender in Research

| | |
|---|---|
| Cell line source(s) | All Trypanosoma brucei brucei Lister 427 cell lines used and generated in this study are described in the methods section. |
| Authentication | RNA-seq provided authentication. |
| Mycoplasma contamination | Mycoplasma contamination check carried our approx. every 3 years - no positive results from those tests to date. |

| Commonly misidentified lines (See ICLAC register) | No commonly misidentified cell lines were used. |
|---|---|

## Plants

| Seed stocks | N/A |
|---|---|
| Novel plant genotypes | N/A |
| Authentication | N/A |

## Flow Cytometry

### Plots

Confirm that:

☒ The axis labels state the marker and fluorochrome used (e.g. CD4-FITC).

☒ The axis scales are clearly visible. Include numbers along axes only for bottom left plot of group (a 'group' is an analysis of identical markers).

☒ All plots are contour plots with outliers or pseudocolor plots.

☒ A numerical value for number of cells or percentage (with statistics) is provided.

### Methodology

| Sample preparation | For single-cell sorting: 5.0 × 10^6 cells were harvested by centrifugation at 4°C and washed twice in sterile filtered ice cold 1X TDB. The cells were resuspended in 1ml of ice cold filtered 1X TDB, and stained with 1μg/ml propidium iodide to exclude dead cells.<br>For VSG-2 expression analysis: 1.0 × 10^6 cells were harvested by centrifugation at 4°C. Cells were incubated in the dark with fluorescently-conjugated anti-VSG-2 diluted 1:500 in HMI-11. Cells were washed three times with 1X TDB and resuspended in 400μl of 1X TDB. |
|---|---|
| Instrument | For VSG expression analysis FACS Canto (BD Biosciences) was used. For single-cell sorting FACS Fusion II cell sorter (BD Biosciences) was used. |
| Software | Data collection (VSG expression analysis) and single-cell sorting were performed with the help of FACSDiva software (BD Biosciences). For VSG expression analysis FlowJo TM Software was used. |
| Cell population abundance | For single-cell sorting the cell population abundance (live cells, singlets) was 11.11%. For VSG-2 expression analysis the cell population abundance (singlets) was between 48.86% and 94.33%, depending on the experiment time point. |
| Gating strategy | For single-cell sorting:<br>Cell populations were gated to remove cellular debris (FSC-A vs SSC-A), doublets (FSC-A vs FSC-H and SSC-A vs SSC-W), and dead cells (positive for propidium iodide staining).<br>For VSG-2 expression analysis:<br>Cell populations were gated to remove cellular debris (FSC-A vs SSC-A) and doublets (FSC-A vs FSC-H). |

☒ Tick this box to confirm that a figure exemplifying the gating strategy is provided in the Supplementary Information.

