## [Peer Review File · Nature]

High-resolution scRNA-seq reveals genomic determinants of VSG expression hierarchy

Corresponding Author: Professor T. Nicolai Siegel

Version 0:

Reviewer comments:

Referee #1

(Remarks to the Author)

Summary of key results: Kirsty R. McWilliam et al. investigate antigenic variation, a common strategy used by pathogens to evade the host immune response, offering intriguing insights into the mechanism behind VSG selection in *Trypanosoma brucei*.

It has been known since 2013 that DNA breaks at fragile subtelomeres can influence the process of antigenic variation in African trypanosomes (Glover, L et al. 2013). In 2023, this idea was applied for the first time when researchers showed that DSBs caused by Cas9 can lead to antigenic variation (Escrivani et al. 2023). Here, the authors have confirmed in their hand that the technique works well to silence a targeted VSG. It is welcome that induction of γ -H2A expression, a hallmark of DSBs, has been added as an internal control.

Originality and significance: The most important difference highlighted in this study concerns the technique chosen to explore the changes in the transcriptome. While Escrivani and coll. (2023) used bulk RNA-seq to investigate VSG expression in a whole population, this study introduces the use of SL-Smart-seq3xpress for single cell RNA sequencing, allowing for a more detailed and individualized analysis.

Data & methodology: Following the successful development and application of the Smart-seq2 method (Müller, et al. Nature, 2018), here they adapt Smart-seq3xpress to *T. brucei*, a scalable method for analyzing multiple samples and low mRNA/ESAG levels that can generate full transcript data to identify VSG isoforms and is accurate enough to separate true biological signals from sequencing noise for VSG analysis. Using the *T. brucei*-tailored SL-Smart-seq3xpress combined with bioinformatic refinement of index reads, this method precisely detects VSG expression in individual cells, significantly reducing background noise. It also allows for the tracking of VSG expression dynamics in single cells before, during, and after the induction of a DSB in VSG-2.

Conclusions: The authors brought interesting data to answer the long-debated question of why specific VSGs dominate during infection. They showed that the availability of homologous DNA for the repair of DSBs and the location of VSG genes within the genome are critical in dictating the hierarchy of VSG activation. Additionally, their study reveals that temporary switching of VSGs might help maintain stable VSG levels when BES1 is being repaired after a break.

Suggested improvements:

1- Page 10: "In stark contrast, when we induced a DSB at the cut-site upstream of the 2nd 70-bp repeats, we observed no switching except in one cell (Fig. 4b). This was despite robust Cas9 induction and elevated γ -H2A signal (Extended Data Fig. 4e,f)."

Could the absence of switching, observed after we induced a DSB at the cut-site upstream of the 2nd 70-bp repeats, be due to Cas9 failing to access the site? Overall, if the γ -H2A signal is detected only in bulk by Western blot, it cannot solely indicate Cas9 activity; off-target effects cannot be ruled out. Additionally, assessing the cutting efficiency is challenging without demonstrating the actual presence of the γ -H2A histone, for instance, through ChIP analysis (see Arnould C et al. Nature 2021). Induced DNA breaks could also be mapped by BLISS (Breaks Labelling In Situ and Sequencing) (Bouwman et al., Nat. Protoc. 15, 3894–3941, 2020).

2- Fig. 3c reveals a significant change in population composition from 4 days to 10 days after a Cas9-induced DSB, with 94% of cells becoming recombinant switchers expressing BES1. However, it would be interesting to plot the intermediate days to determine whether this transition occurs gradually.

3- Fig. 3c and Lines 312-317: Could it be that BES1 is maintained in a chromatin state that is poised, enabling reversible changes in its expression, or do they believe it transitions from an active to a silenced state through heterochromatin formation? Have you considered evaluating chromatin accessibility, perhaps using ATAC-seq, to provide deeper insight into these changes before and after switching? Such an analysis could add valuable detail to your findings.

Minor points:

- Please define "SL" at its first occurrence in the manuscript (page 2). Ensuring that all acronyms are explained when they first appear will aid in reader comprehension.
- MES" (metacyclic expression sites) should also be defined at its initial mention.
- Regarding the order of figures, it appears Fig 3d is referenced before Fig 3c.

Referee #2

(Remarks to the Author)

The paper by McWilliam et al deploys an excellent new tool with which it describes in fine detail what has already been understood from previous literature (where the tools used were far more blunt). Aside from validating what was already known, the authors make two truly novel observations : (a) that the best studied VSG (VSG2) is in fact expressed by the most stable (for some reason) BES (BES1), (b) that this VSG (VSG2) is also unique - and this has specific consequences for recombinational repair of a DNA break. Moving from VSG2 to VSG8 (with multiple homologous copies within the archive) selects for what sounds like mosaics (extended figure 5).

However there is no mechanistic insight for novel observation (a), leaving novel observation (b) as the crux of the paper. For this to be considered for Nature I would propose that the paper is entirely reorganized around novel observation (b). The rest of it is fine, but it's really not novel (but rather the adaptation of cutting edge single cell tools to parasitology). Thus Figure 2 can be moved to supplement, Figure 3 can be consolidated with Figure 4 (if needed, as both offer fine tuning of older observations)- and extended Fig 5 should really be moved to the body of the manuscript (it is the key novelty here).

Minor comments:

Lines 275-276:

"Given that these VSGs are amongst the largest in the genome, this observation suggests that VSG growth dynamics are not primarily governed by VSG length, as previously suggested".

This is a point well taken although the experiment in question (ref18) is a modeling experiment validated by in vivo infection data. I would like to see this statement tempered to reflect this fact (because here it is entirely in vitro data we are discussing).

Lines 316-317

"However, these variants appeared either to have been outcompeted by cells that had maintained BES1 expression or to have switched back to transcribing BES1 once the DSB was repaired."

How? what would the mechanism for that be? Perhaps something previously suggested by PMID: 19915072? This can be expanded upon, a bit. (This also highlights the fascination with technology development evident throughout the paper - rather than the information yielded by the technology)

lines 307-311

"Interestingly, 10 days after Cas9 induction, we observed a surprising shift in the population composition compared to the 96 hours time point. Now 94.1% of the cells were recombinant switchers transcribing BES1, suggesting that there is a strong fitness advantage in keeping BES1 active (Fig. 3c). S

How? What is unique about BES1? (this is point (a) in my review - important to dig into.

Version 1:

Reviewer comments:

Referee #1

(Remarks to the Author)

The authors have thoroughly addressed all reviewer comments, introducing substantial revisions and additional data, such

as BLISS and ATAC-seq analyses, that significantly strengthen the manuscript. The sharpened focus on parasite biology over technology enhances the study's relevance, making it a valuable and impactful contribution to parasitology and the broader field of epigenetics.

Referee #2

(Remarks to the Author)

In the rebuttal it is stated that "the presence or absence of a suitable homologous repair template determines how a DSB is repaired in the active VSG...is the main conclusion of this study".

If this is indeed the case, then it is an incremental advance or rather a starting hypothesis to the identification of mechanism. Clearly, building toward mechanism must start from an observation - but one would argue that the observation (that you need the presence or absence of a repair template) as stated by the authors sounds like pre-existing knowledge.

The paper is a lot more than that (the rebuttal shortchanges it). And the comprehensive revision was highly responsive to both reviewers' comments. It will be a strong technical addition to the literature (with the beginnings of exciting new biology as well).

Response to reviewers

We thank the reviewers for their positive feedback and constructive criticism and have responded to each of their comments in the below document.

Referee #1 (Remarks to the Author):

Summary of key results: Kirsty R. McWilliam et al. investigate antigenic variation, a common strategy used by pathogens to evade the host immune response, offering intriguing insights into the mechanism behind VSG selection in *Trypanosoma brucei*.

It has been known since 2013 that DNA breaks at fragile subtelomeres can influence the process of antigenic variation in African trypanosomes (Glover, L et al. 2013). In 2023, this idea was applied for the first time when researchers showed that DSBs caused by Cas9 can lead to antigenic variation (Escrivani et al. 2023). Here, the authors have confirmed in their hand that the technique works well to silence a targeted VSG. It is welcome that induction of γ -H2A expression, a hallmark of DSBs, has been added as an internal control.

Originality and significance: The most important difference highlighted in this study concerns the technique chosen to explore the changes in the transcriptome. While Escrivani and coll. (2023) used bulk RNA-seq to investigate VSG expression in a whole population, this study introduces the use of SL-Smart-seq3xpress for single cell RNA sequencing, allowing for a more detailed and individualized analysis.

Data & methodology: Following the successful development and application of the Smart-seq2 method (Müller, et al. Nature, 2018), here they adapt Smart-seq3xpress to *T. brucei*, a scalable method for analyzing multiple samples and low mRNA/ESAG levels that can generate full transcript data to identify VSG isoforms and is accurate enough to separate true biological signals from sequencing noise for VSG analysis. Using the *T. brucei*-tailored SL-Smart-seq3xpress combined with bioinformatic refinement of index reads, this method precisely detects VSG expression in individual cells, significantly reducing background noise. It also allows for the tracking of VSG expression dynamics in single cells before, during, and after the induction of a DSB in VSG-2.

Conclusions: The authors brought interesting data to answer the long-debated question of why specific VSGs dominate during infection. They showed that the availability of homologous DNA for the repair of DSBs and the location of VSG genes within the genome are critical in dictating the hierarchy of VSG activation. Additionally, their study reveals that temporary switching of VSGs might help maintain stable VSG levels when BES1 is being repaired after a break.

Suggested improvements:

1- Page 10: "In stark contrast, when we induced a DSB at the cut-site upstream of the 2nd 70-bp repeats, we observed no switching except in one cell (Fig. 4b). This was despite robust Cas9 induction and elevated γ -H2A signal (Extended Data Fig. 4e,f)."

Could the absence of switching, observed after we induced a DSB at the cut-site upstream of the 2nd 70-bp repeats, be due to Cas9 failing to access the site? Overall, if the γ -H2A signal is detected only in bulk by Western blot, it cannot solely indicate Cas9 activity; off-target effects cannot be ruled out. Additionally, assessing the cutting efficiency is challenging without demonstrating the actual presence of the γ -H2A histone, for instance, through ChIP analysis (see Arnould C et al. Nature 2021). Induced DNA breaks could also be mapped by BLISS (Breaks Labelling In Situ and Sequencing) (Bouwman et al., Nat. Protoc. 15, 3894–3941, 2020).

This is a very good point. The absence of switching could well be caused by Cas9 not being able to access the site, and we recognize the limitations of performing a western blot to detect γ -H2A signal, as this could be caused by off-target effects. To address the reviewer's concerns, we have performed BLISS in duplicate to map the site of the induced DSB in the following cell lines mentioned in the originally submitted manuscript:

- BES1.40225 (BES1 between ESAG 3 and 8). No VSG switching.
- BES1.54824 (BES1 between 70bp repeats and VSG-2). VSG switching
- VSG2.152 (VSG-2 CDS). VSG switching.
- VSG2.782 (VSG-2 CDS). VSG switching.
- BES1.58149 (telomere adjacent on BES1). No VSG switching.

- VSG8.609 (VSG-8 CDS). VSG switching
- VSG8.1105 (VSG-8 CDS). VSG switching

We observed a clean BLISS signal for the four cell lines in which Cas9 induction resulted in VSG switching. We have added the BLISS data to Fig. 1c, Extended Data Fig. 4c and Extended Data Fig. 6c.

No BLISS signal was observed for the two cell lines in which Cas9 induction did not result in VSG switching (BES1.40225 and BES1.58149). Thus, the absence of switching in these cell lines may well be due to a failure to efficiently induce a DSB, rather than being position dependent. Alternatively, it is possible that the dynamics of DSB induction are different for these two cell lines, as the γ -H2A western blot was performed 24 h after Cas9 induction and the BLISS assays were performed 4 h after Cas9 induction.

As reviewer 2 had suggested to restructure the manuscript and to focus on our novel observation, the importance of a suitable repair template, we decided not to pursue the γ -H2A/BLISS discrepancy and to remove any reference to attempted DSB inductions upstream of the 70 bp repeat and downstream of the VSG that did not result in VSG switching.

In three of the four cell lines in which Cas9 induction resulted in a switch in VSG expression, the dominant BLISS site was at the expected position. The exception was cell line VSG2.152 (VSG-2 CDS). Here we observed the peak approximately 890 bp downstream of the expected site, still within the VSG ORF. Sanger sequencing of the integrated sgRNA DNA revealed that the shift of the BLISS peak was not caused by a

Cas9 off-target effect, but by transfection of a different sgRNA DNA template. The BLISS peak perfectly matched the integrated sgRNA DNA sequence.

We suspect that the reason for the discrepancy is that we originally designed many different sgRNA and mislabelled two of them. Therefore, VSG2.152 (VSG-2 CDS) should be renamed to VSG2.1140. While this is obviously very unfortunate and should not have happened, this error does not affect the results and conclusions of our study. The VSG2.1140 break position is still within the VSG-2 CDS and has no homology to other VSG sequences, and thus our main findings, that the presence or absence of a suitable homologous repair template, as well as the location of the new VSG gene, determines the DSB repair mechanism and the frequency with which a specific VSG gene is activated, remain true. In addition, we verified all other cell lines used in this study by Sanger sequencing and confirmed that their sgRNAs map to the intended site.

Overall, these experiments demonstrate the importance of confirming expected Cas9 DSB induction using BLISS and we are glad that the reviewer suggested this control.

2- Fig. 3c reveals a significant change in population composition from 4 days to 10 days after a Cas9-induced DSB, with 94% of cells becoming recombinant switchers expressing BES1. However, it would be interesting to plot the intermediate days to determine whether this transition occurs gradually.

As suggested by the reviewer, we have repeated the switching experiments to obtain the ratio between transcriptional switchers and recombination switchers for additional time points. We have now determined the transcriptomes of an additional ~4,500 cells at 0, 3, 4, 6 and 10 days after DSB induction in VSG-2 at position VSG2.1140. Because at day 6, most transcriptional switchers had disappeared, we decided not to sequence cells collected at day 8.

In addition, we determined the ratio of transcriptional/recombinational switchers at day 3 of our initial VSG switching time course. The new results have been added to Figure 3e and together show a rapid loss of transcriptional switchers. We do also see a small temporal shift between the two separate time courses (but not between the two replicates of each time course). We suspect that these differences are due to slight differences in the time between DSB induction with doxycycline and cell harvest for sorting, or slightly different parasite concentrations in the two starting cultures.

Nevertheless, we believe that the new data show that the loss of transcriptional switchers is very rapid, making it somewhat unlikely that they are simply lost by overgrowth.

3- Fig. 3c and Lines 312-317: Could it be that BES1 is maintained in a chromatin state that is poised, enabling reversible changes in its expression, or do they believe it transitions from an active to a silenced state through heterochromatin formation? Have you considered evaluating chromatin accessibility, perhaps using ATAC-seq, to provide deeper insight into these changes before and after switching? Such an analysis could add valuable detail to your findings.

Although 'classical' heterochromatin marks such as H3K9me3 and HP1 appear to be absent in *T. brucei*, both scenarios seem plausible. However, we prefer a model in which the transcriptional silencing of BES1 is a direct consequence of the DSB in the VSG. We speculate that, as in other organisms, DSBs in the vicinity of transcriptionally active genes trigger transient transcriptional silencing through complex multistep cascades involving different proteins and that the previously active BES remains 'poised' for reactivation once the DSB is repaired. Indeed, it has been suggested that the previously active BES remains poised for reexpression following a transcriptional switch (Aresta-Branco et al., 2016, PMID: 26673706).

We agree with the reviewer that ATAC-seq would be well suited to address this question and performed ATAC-seq as suggested. We used a cell line containing a puromycin drug resistance gene in BES1 (containing VSG-2) and a neomycin drug resistance gene in BES17 (containing VSG-13)(Figueiredo & Cross, 2010, PMID: 19915072). By changing the drug selection from puromycin to neomycin, we can select for cells that have switched from BES1 to BES17. Using this cell line, we performed ATAC-seq and RNA-seq on six bulk cultures of cells transcribing BES1 or BES17 before the drug switch, right after these cultures recovered from a drug switch (day 5 or day 7, depending on the direction of the switch), and 14 days after a drug switch. We added our results to the manuscript, lines 289-302:

ATAC-seq assays performed 0, 7 and 14 days after replacing puromycin with neomycin to activate BES17 with *VSG-13*, indicated that even at day 14 BES1 was more open than the silent BESs (Fig. 3f). We also observed that *VSG-2* transcript levels were only partially reduced (Fig. 3f). Since the experiment was performed in bulk, we cannot say for certain that BES1 stayed open and 'poised' for reactivation- whilst BES1 may have stayed poised in some cells, in other cells BES1 may have never stopped being transcribed.

Interestingly however, when performing the reverse experiment (replacing neomycin with puromycin to activate BES1 with *VSG-2*), we found that *VSG-13* transcript levels decreased rapidly (Fig. 3g). In addition, ATAC-seq data indicated that BES17 was closed by day 14. Thus, while these assays cannot rule out or confirm that a BES stays poised after a transcriptional switch, they indicate that a switch away from BES1 follows different dynamics than a switch from BES 17 to BES1, again suggesting that there may be an advantage to expressing BES1.

Minor points:

- Please define "SL" at its first occurrence in the manuscript (page 2). Ensuring that all acronyms are explained when they first appear will aid in reader comprehension.

SL has now been defined at its first occurrence (lines 92-93).

- "MES" (metacyclic expression sites) should also be defined at its initial mention.

MES has now been defined at its first occurrence (line 226).

- Regarding the order of figures, it appears Fig 3d is referenced before Fig 3c.

All figures have now been checked to ensure their ordering is correct.

Referee #2 (Remarks to the Author):

The paper by McWilliam et al deploys an excellent new tool with which it describes in fine detail what has already been understood from previous literature (where the tools used were far more blunt). Aside from validating what was already known, the authors make two truly novel observations : (a) that the best studied VSG (VSG2) is in fact expressed by the most stable (for some reason) BES (BES1), (b) that this VSG (VSG2) is also unique - and this has specific consequences for recombinational repair of a DNA break. Moving from VSG2 to VSG8 (with multiple homologous copies within the archive) selects for what sounds like mosaics (extended figure 5).

However there is no mechanistic insight for novel observation (a), leaving novel observation (b) as the crux of the paper. For this to be considered for Nature I would propose that the paper is entirely reorganized around novel observation (b).

The rest of it is fine, but it's really not novel (but rather the adaptation of cutting edge single cell tools to parasitology). Thus Figure 2 can be moved to supplement, Figure 3 can be consolidated with Figure 4 (if needed, as both offer fine tuning of older observations)- and extended Fig 5 should really be moved to the body of the manuscript (it is the key novelty here).

We agree with the comments regarding the focus of our manuscript and have now restructured the manuscript to focus more on the novel observation (b). In this regard, we have moved the data for the additional breakpoints VSG2.782 and BES1.54824 to Extended Data Figures 4 and 5 and removed the data for DSBs upstream of the 70 bp repeat region and downstream of the VSG. Fig. 5 has now become Fig. 4, and we have added the data from Extended Data Fig. 5c to the body of the manuscript (Fig. 4) as suggested. In addition, we have added ATAC-seq and RNA-seq data suggesting that a switch from BES17 to BES1 may be faster than a switch in the opposite direction, again suggesting that there may be some advantage for cells to transcribe BES1 under *in vitro* conditions.

Regarding the comments on method development, we accept that there is a strong technology focus in this manuscript. However, since SL-Smart-seq3xpress is among the most sensitive single cell methods described (>80% of total cellular transcripts detected) and will be an important tool for the field given its ability to link transcriptomic changes to underlying genomic changes, we believe this focus is justified. In addition, we believe it is important to highlight the risk that scRNA-seq data artifacts may pose to the study of mutually exclusive expressed genes. An apparent loss of mutually exclusive expression may simply be the result of scRNA-seq sequencing or analysis artifacts (Fig. 2d). Therefore, we have retained some of the panels from Figure 2 in the body of the manuscript.

Minor comments:

Lines 275-276:

"Given that these VSGs are amongst the largest in the genome, this observation suggests that VSG growth dynamics are not primarily governed by VSG length, as previously suggested".

This is a point well taken although the experiment in question (ref18) is a modeling experiment validated by *in vivo* infection data. I would like to see this statement tempered to reflect this fact (because here it is entirely *in vitro* data we are discussing).

We have now changed our statement to clarify that our conclusion is based on *in vitro* observations. Lines 234-236 "Given that these VSGs are amongst the largest in the genome, this observation suggests that, *in vitro*, VSG growth dynamics are not primarily governed by VSG length".

Lines 316-317

"However, these variants appeared either to have been outcompeted by cells that had maintained BES1 expression or to have switched back to transcribing BES1 once the DSB was repaired."

How? what would the mechanism for that be? Perhaps something previously suggested by PMID: 19915072? This can be expanded upon, a bit. (This also highlights the fascination with technology development evident throughout the paper - rather than the information yielded by the technology)

Yes, as suggested in the publication by Figueiredo & Cross (PMID: 19915072), we also speculated that while BES1 undergoes repair after the DSB, it remains in an 'open' conformation and is ready for re-expression after repair.

To further investigate this possibility, we performed ATAC-seq experiments as suggested by reviewer 1 to assess the openness of the BES during a transcriptional switch. We used a cell line containing a puromycin drug resistance gene in BES1 (containing *VSG-2*) and a neomycin drug resistance gene in BES17 (containing *VSG-13*) (cell line BF-LF17.13 in Figueiredo & Cross, 2010, PMID: 19915072). By changing the drug selection from puromycin to neomycin, we can select for cells that have switched from BES1 to BES17. Using this cell line, we performed ATAC-seq and RNA-seq on six bulk cultures of cells transcribing BES1 or BES17 before the switch in drug pressure, immediately after these cultures recovered from a drug switch (day 5 or day 7, depending on the direction of the switch), and 14 days after a drug switch. We have included our results in the manuscript, lines 289-302:

ATAC-seq assays performed 0, 7 and 14 days after replacing puromycin with neomycin to activate BES17 with *VSG-13*, indicated that even at day 14 BES1 was more open than the silent BESs (Fig. 3f). We also observed that *VSG-2* transcript levels were only partially reduced (Fig. 3f). Since the experiment was performed in bulk, we cannot say for certain that BES1 stayed open and 'poised' for reactivation- whilst BES1 may have stayed poised in some cells, in other cells BES1 may have never stopped being transcribed.

Interestingly however, when performing the reverse experiment (replacing neomycin with puromycin to activate BES1 with *VSG-2*), we found that *VSG-13* transcript levels decreased rapidly (Fig. 3g). In addition, ATAC-seq data indicated that BES17 was closed by day 14. Thus, while these assays cannot rule out or confirm that a BES stays poised after a transcriptional switch, they indicate that a switch away from BES1 follows different dynamics than a switch from BES 17 to BES1, again suggesting that there may be an advantage to expressing BES1.

lines 307-311

"Interestingly, 10 days after Cas9 induction, we observed a surprising shift in the population composition compared to the 96 hours time point. Now 94.1% of the cells were recombinant switchers transcribing BES1, suggesting that there is a strong fitness advantage in keeping BES1 active (Fig. 3c). S

How? What is unique about BES1? (this is point (a) in my review - important to dig into.

The widespread use of BES1 by *T. brucei* under cell culture conditions has intrigued researchers for years, and so we agree that the loss of transcriptional switchers over time is an interesting observation. By adding additional time points to our switch time course analysis and by determining the openness of the BESs during a switch, as suggested by reviewer 1, we have tried to shed some light on this point.

A growth advantage of cells transcribing BES1 could be conferred by one or more of the ESAGs located in this BES. Although the order of the ESAGs on the BESs remains largely the same, not all BESs contain the full repertoire of ESAGs. Thus, variation between expressed ESAGs could confer growth advantages. We have restructured the manuscript as suggested. However, we believe that deciphering exactly how BES1 confers a growth advantage is beyond the scope of this study and does not directly relate to the main conclusion of this study, i.e. that the presence or absence of a suitable homologous repair template determines how a DSB is repaired in the active VSG.